# DUBStepR is a scalable correlation-based feature selection method for accurately clustering single-cell data

Bobby Ranjan [1], Wenjie Sun[1], Jinyu Park[1], Kunal Mishra[1], Florian Schmidt[1], Ronald Xie[1], Fatemeh Alipour[1], Vipul Singhal[1], Ignasius Joanito [1], Mohammad Amin Honardoost[1,2], Jacy Mei Yun Yong[3], Ee Tzun Koh[3], Khai Pang Leong[3], Nirmala Arul Rayan[1], Michelle Gek Liang Lim[1] & Shyam Prabhakar [1✉]

Feature selection (marker gene selection) is widely believed to improve clustering accuracy, and is thus a key component of single cell clustering pipelines. Existing feature selection methods perform inconsistently across datasets, occasionally even resulting in poorer clustering accuracy than without feature selection. Moreover, existing methods ignore information contained in gene-gene correlations. Here, we introduce DUBStepR (Determining the Underlying Basis using Stepwise Regression), a feature selection algorithm that leverages gene-gene correlations with a novel measure of inhomogeneity in feature space, termed the Density Index (DI). Despite selecting a relatively small number of genes, DUBStepR substantially outperformed existing single-cell feature selection methods across diverse clustering benchmarks. Additionally, DUBStepR was the only method to robustly deconvolve T and NK heterogeneity by identifying disease-associated common and rare cell types and subtypes in PBMCs from rheumatoid arthritis patients. DUBStepR is scalable to over a million cells, and can be straightforwardly applied to other data types such as single-cell ATAC-seq. We propose DUBStepR as a general-purpose feature selection solution for accurately clustering single-cell data.

[1] Laboratory of Systems Biology and Data Analytics, Genome Institute of Singapore, A*STAR, 60 Biopolis Street, Singapore 138672, Singapore. [2] Department of Medicine, School of Medicine, National University of Singapore, 21 Lower Kent Ridge Road, Singapore 119077, Singapore. [3] Department of Rheumatology, Allergy and Immunology, Tan Tock Seng Hospital, Singapore 308433, Singapore. ✉email: prabhakars@gis.a-star.edu.sg

Heterogeneity in single-cell RNA sequencing (scRNA-seq) datasets is frequently characterized by identifying cell clusters in gene expression space, wherein each cluster represents a distinct cell type or cell state. In particular, numerous studies have used unsupervised clustering to discover novel cell populations in heterogeneous samples[1]. The steps involved in unsupervised clustering of scRNA-seq data have been well documented[2]. (i) Low-quality cells are first discarded in a quality control step[3]. (ii) Reads obtained from the remaining cells are then normalized to remove the influence of technical effects, while preserving true biological variation[4]. (iii) After normalization, feature selection is performed to select the subset of genes that are informative for clustering, (iv) which are then typically reduced to a small number of dimensions using principal component analysis (PCA)[5]. (v) In the reduced principal component (PC) space, cells are clustered based on their distance from one another (typically, Euclidean distance), and (vi) the corresponding clusters are assigned a cell type or state label based on the known functions of their differentially expressed (DE) genes[6].

Although feature selection is a critical step in the canonical clustering workflow described above, only a few different approaches have been developed in this space. Moreover, there have been only a handful of systematic benchmarking studies of scRNA-seq feature selection methods[7–9]. A good feature selection algorithm is one that selects cell-type-specific (DE) genes as features, and rejects the remaining genes. More importantly, the algorithm should select features that optimize the separation between biologically distinct cell clusters. A comprehensive benchmarking study of feature selection methods would ideally use both of these metrics.

The most widely used approach for feature selection is mean-variance modeling: genes whose variation across cells exceeds a data-derived null model are selected as features[10,11]. Such genes are described as highly variable genes (HVGs)[12]. Some earlier single-cell studies instead selected genes with high loading on the top principal components of the gene expression matrix (high loading genes, or HLGs) as features[13]. M3Drop, a more recent method, selects genes whose dropout rate (number of cells in which the gene is undetected) exceeds that of other genes with the same mean expression[9]. As an alternative approach to detect rare cell types, GiniClust uses a modified Gini index to identify genes whose expression is concentrated in a relatively small number of cells[14]. All of the above feature selection methods test genes individually, without considering expression relationships between genes. Another drawback is that existing methods for determining the size of the feature set do not bear direct relation to the separation of cells in the resulting space.

Here, we present Determining the Underlying Basis using Stepwise Regression (DUBStepR), an algorithm for feature selection based on gene–gene correlations. A key feature of DUBStepR is the use of a stepwise approach to identify an initial core set of genes that most strongly represent coherent expression variation in the dataset. Uniquely, DUBStepR defines a novel graph-based measure of cell aggregation in the feature space (termed density index (DI)), and uses this measure to optimize the number of features. The complete DUBStepR workflow is shown in Fig. 1.

We benchmarked DUBStepR against 7 commonly used feature selection algorithms on datasets from four different scRNA-seq protocols (10x Genomics, Drop-Seq, CEL-Seq2, and Smart-Seq2) and found that it substantially outperformed other methods on quantitative measures of cluster separation and marker gene detection. Further, DUBStepR uniquely deconvolved T and NK cell heterogeneity by identifying disease-pertinent clusters (of both rare and common abundances) in PBMCs from rheumatoid arthritis patients. Finally, we show that DUBStepR could potentially be applied even to single-cell ATAC sequencing data.

## Results

**Gene–gene correlations predict cell-type-specific DE genes.** The first step in DUBStepR is to select an initial set of candidate features based on known properties of cell-type-specific DE genes (marker genes). DE genes specific to the same cell types would tend to be highly correlated with each other, whereas those specific to distinct cell types are likely to be anti-correlated (Fig. 2a, b; see the "Methods" section). In contrast, non-DE genes are likely to be only weakly correlated (Fig. 2c). We therefore hypothesized that a correlation range score derived from the difference between the strongest positive and strongest negative correlation coefficients of a gene ("Methods"), would be substantially elevated among DE genes. Indeed, we found that the correlation range score was significantly higher for DE genes relative to non-DE genes (Fig. 2d). Moreover, the correlation range score of a gene was highly predictive of its greatest fold change between cell types, and also its most significant differential expression $q$-value (Fig. 2e, f). Due to the strong association between correlation range and marker gene status, DUBStepR selects genes with high correlation range score as the initial set of candidate feature genes ("Methods").

**Stepwise regression identifies a minimally redundant feature subset.** We observed that candidate feature genes formed correlated blocks of varying size in the gene–gene correlation (GGC) matrix (Fig. 3a), with each block presumably representing a distinct pattern of expression variation across the cells. To ensure a more even representation of the diverse cell-type-specific expression signatures within the candidate feature set, we sought to identify a representative minimally redundant subset, which we termed "seed" genes. For this purpose, DUBStepR performs stepwise regression on the GGC matrix, regressing out, at each step, the gene explaining the largest amount of variance in the residual from the previous step (Fig. 3b–d). We devised an efficient implementation of this procedure that requires only a single matrix multiplication at each step ("Methods").

This approach selects seed genes with diverse patterns of cell-type-specificity (Fig. 3e–h). DUBStepR then uses the elbow point of the stepwise regression scree plot to determine the optimal number of steps ("Methods"), i.e., the size of the seed gene set (Fig. 3i, j).

**Guilt-by-association expands the feature set.** Although the seed genes in principle span the major expression signatures in the dataset, each individual signature (set of correlated genes) is now represented by only a handful of genes (2–5 genes, in most cases). Given the high level of noise in scRNA-seq data, it is likely that this is insufficient to fully capture coherent variation across cells. DUBStepR therefore expands the seed gene set by iteratively adding correlated genes from the candidate feature set using a guilt-by-association approach. Guilt-by-association has previously been employed for feature selection on mass spectrometry data[15], and provides a robust solution to order candidate feature genes by their association to the seed gene set (Supp. Fig. S2; "Methods"). This approach allows DUBStepR to prioritize genes that more strongly represent an expression signature (i.e., are better features for clustering). Candidate genes are added until DUBStepR reaches the optimal number of feature genes (see below).

**Benchmarking.** To benchmark the performance of DUBStepR, we compared it against 6 other algorithms for feature selection in scRNA-seq data: three variants of the HVG approach (HVGDisp, HVGVST, trendVar), deviance-based feature selection (devianceFS), HLG, and M3Drop/DANB (Table 1). For completeness, we

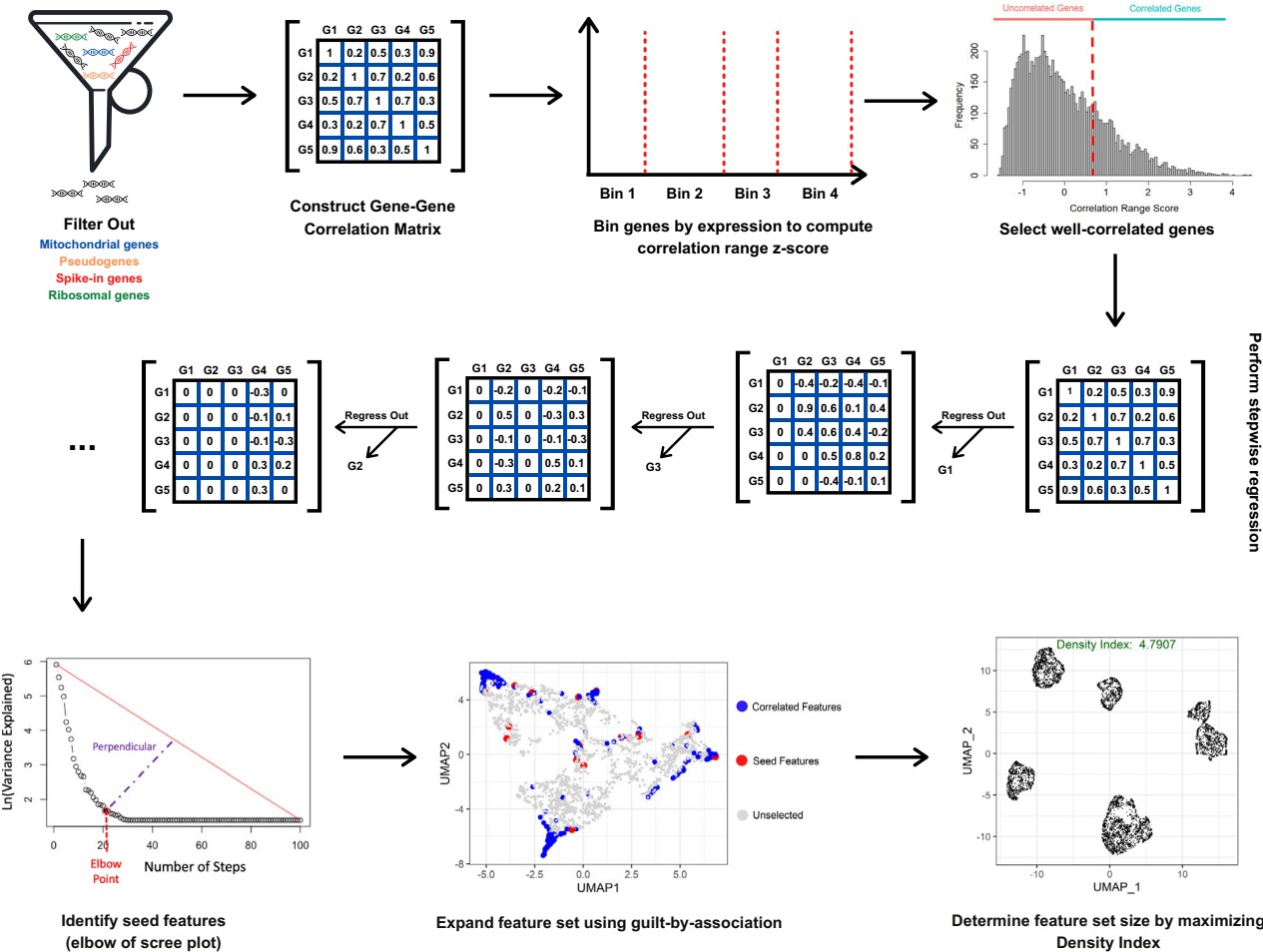

**Fig. 1 Overview of DUBStepR workflow.** After filtering out mitochondrial, ribosomal, spike-in, and pseudogenes, DUBStepR constructs a GGC matrix and bins genes by expression to compute their correlation range z-scores, which are used to select well-correlated genes. DUBStepR then performs stepwise regression on the GGC matrix to identify a minimally redundant subset of seed features, which are then expanded by adding correlated features (guilt-by-association). The optimal feature set size is determined using the Density Index metric.

also benchmarked GiniClust, though it was designed only for identifying markers of rare cell types. Each algorithm was benchmarked on 7 datasets spanning 4 scRNA-seq protocols: 10x Genomics, Drop-Seq, CEL-Seq2, and Smart-Seq on the Fluidigm C1 (Supp. Note 2A). These datasets were selected because the true cell type could be independently ascertained based on cell line identity or FACS gate. Our benchmarking approach thus avoids the circularity of using algorithmically defined cell type labels as ground truth.

To evaluate the quality of the selected features, we used the well-established Silhouette index (SI), which quantifies cluster separation, i.e., closeness between cells belonging to the same cluster, relative to the distance to cells from other clusters[16] (Supp. Note 3A). In addition to being a well-established measure of single-cell cluster separation[17–19], the SI has the advantage of being independent of any downstream clustering algorithm. We evaluated the SI of each algorithm across a range of feature set sizes (50–4000), scaled the SI values to a maximum of 1 for each dataset, and then averaged the scaled SIs across the 7 datasets (Fig. 4a; Supp. Fig. S3). Remarkably, HLG, an elementary PCA-based method that predates scRNA-seq technology, achieved greater average cell type separation than existing single-cell algorithms at most feature set sizes. In contrast to DUBStepR, which showed maximal performance at 200–300 features, the other methods remained close to their respective performance

peaks over a broad range from 300 to 2000 features and dropped off on either side of this range. DUBStepR substantially outperformed all other methods across the entire range of feature set size (Fig. 4a). Moreover, DUBStepR was the top-ranked algorithm on 5 of the 7 datasets (Fig. 4b).

For optimal cell type clustering, a feature selection algorithm should ideally select only DE genes, i.e., genes specific to cell types or subtypes, as features. As an independent benchmark, we therefore quantified the ability of feature selection algorithms to discriminate between DE and non-DE genes. To minimize the effect of ambiguously classified genes, we designated the top 500 most differentially expressed genes in each dataset as DE, and the bottom 500 as non-DE ("Methods"), and then quantified performance using the area under the receiver operating characteristic (AUROC; Supp. Note 3B). Remarkably, DUBStepR achieved an AUROC in excess of 0.97 on all 7 datasets, indicating near-perfect separation of DE and non-DE genes (Fig. 4c). devianceFS was able to exceed the same performance threshold on 4 of the 7 datasets and HLG on only one. All other methods demonstrated significantly lower performance (Fig. 4c). Thus, DUBStepR greatly improves our ability to select cell type/ subtype-specific marker genes (DE genes) for clustering scRNA-seq data.

With the exponential increase in the size of single-cell datasets, any new computational approach in the field must be

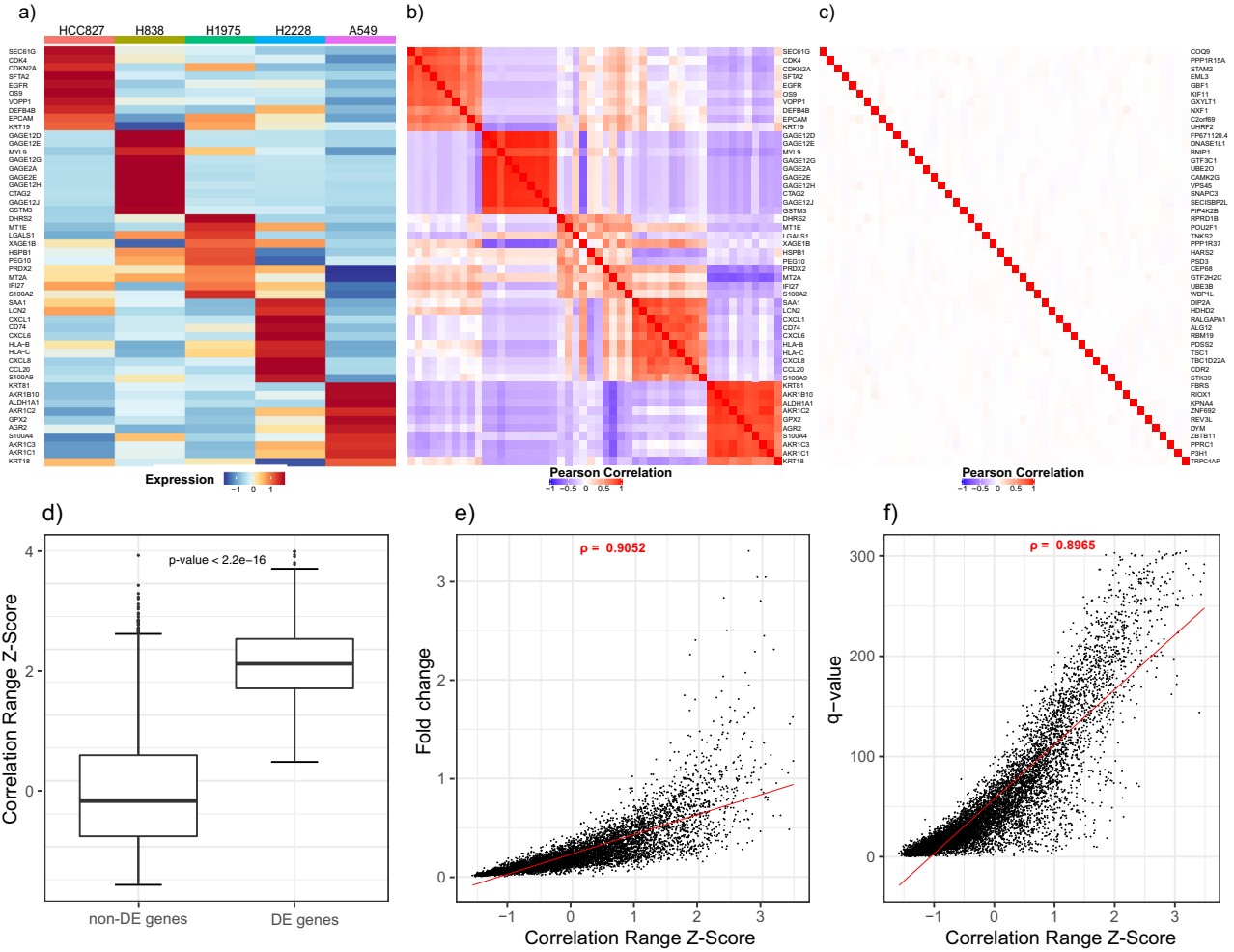

**Fig. 2 Expression correlations of DE genes: scRNA-seq data from five lung adenocarcinoma cell lines[28].** **a** Average expression of top 10 DE genes for each cell line. **b** Gene–gene correlations of the same genes. **c** Gene–gene correlations for non-DE genes. **d** Boxplot showing correlation range scores for non-DE and DE genes. DE gene computation is detailed in Supp. Note 1A. The middle line is the median, the lower and upper box limits correspond to the first and third quartiles, the upper and lower whiskers extend upto 1.5*IQR from the top and bottom of the box, respectively (where IQR is the inter-quartile range), and data beyond the ends of the whiskers are outlying points that are plotted individually. *P*-value was calculated using two-sided Wilcoxon test (*n* = 8602 expressed genes). **e**, **f** Scatter plot of genes showing correlation between **e** log2(fold change) of cell-type-specific expression and **f** −log10(*q*-value) of cell-type-specific expression with correlation range score. $\rho$: Spearman correlation.

able to scale to over a million cells. To improve DUBStepR's ability to efficiently process large datasets, we identified a technique to reduce a key step in stepwise regression to a single matrix multiplication, sped up calculation of the elbow point, and implemented the entire workflow on sparse matrices ("Methods"). To benchmark scalability, we profiled execution time and memory consumption of DUBStepR, as well as the other aforementioned feature selection methods, on a recent mouse organogenesis dataset of over 1 million cells[20]. This dataset was downsampled to produce two additional datasets of 10*k* and 100*k* cells, respectively, while maintaining cell-type diversity (Supp. Note 2B). DUBStepR, HVGDisp, HVGVST, trendVar, devianceFS, and M3DropDANB were able to process the entire 1 million cell dataset, while GiniClust and HLG could not scale to 100*k* cells (Supp. Fig. S4). On the largest dataset, DUBStepR ranked fourth out of the eight tested methods in memory consumption and compute time. In terms of memory scalability, DUBStepR used 6.4x more memory to process the 1M cell dataset as compared to the 100*k* dataset. In contrast, HVGDisp, HVGVST, trendVar, devianceFS, and M3Drop-DANB all increased their memory consumption by 12.5x.

Thus, DUBStepR is scalable to over a million cells and shows promise for even larger datasets.

**Density index predicts the optimal feature set.** As shown above, selecting too few or too many feature genes can result in sub-optimal clustering (Fig. 4a). Ideally, we would want to select the feature set size that maximized cell type separation (i.e., the SI) in the feature space. However, since the feature selection algorithm by definition does not know the true cell type labels, it is not possible to calculate the SI for any given feature set size. We therefore endeavored to define a proxy metric that would approximately model the SI without requiring knowledge of cell-type labels. To this end, we defined a measure of the inhomogeneity or "clumpiness" of the distribution of cells in feature space. If each cell clump represented a distinct cell type, then this measure would tend to correlate with the SI. The measure, which we termed the density index (DI), equals the root-mean squared distance between all cell pairs, divided by the mean distance between a cell and its *k* nearest neighbors ("Methods"). Intuitively, when cells are well clustered and therefore inhomogeneously distributed in feature space, the distance to nearest

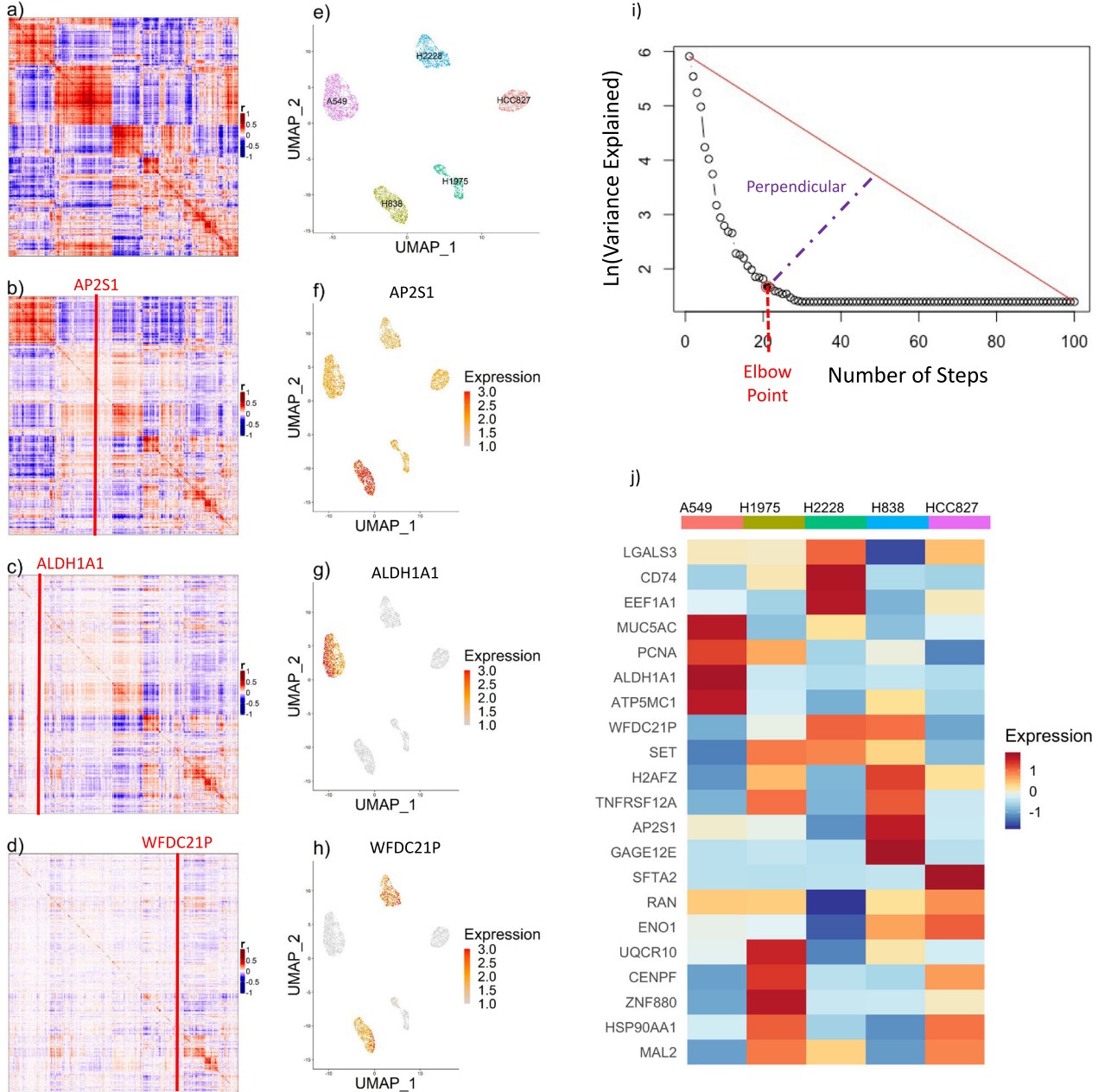

**Fig. 3 Stepwise regression to identify a minimally redundant feature subset. a** Gene–gene correlation matrix of candidate feature genes (high correlation range score). **b–d** Residuals from stepwise regression on the gene–gene correlation matrix. **e** UMAP visualization of cells in an optimal feature space, colored by cell line. **f–h** Same UMAP, colored by expression of genes regressed out in the first three steps. **i** Scree plot: variance in GGC matrix explained by the gene regressed out at each step. **j** Standardized average expression of the final seed gene set in each of the 5 cell lines. r: Pearson correlation.

| Table 1 Feature selection methods used for the benchmarking comparison. | | |
|---|---|---|
| **Algorithm** | **Description** | **Software** |
| devianceFS | Feature selection by approximate multinomial deviance | scry[39,40] |
| GiniClust | Gini index-based feature selection | GiniClust[14] |
| HLG | Features ranked by the sum of magnitude of PC loadings | irlba[9,41] |
| HVGDisp | Highly variable genes by dispersion value | Seurat[10] |
| HVGVST | Highly variable genes after variance-stabilized transformation | Seurat[42] |
| M3Drop/DANB | Dropout-based feature selection: M3Drop for read counts, DANB for UMIs | M3Drop[9] |
| trendVar | Biological and technical components of the gene-specific variance | scran[11] |

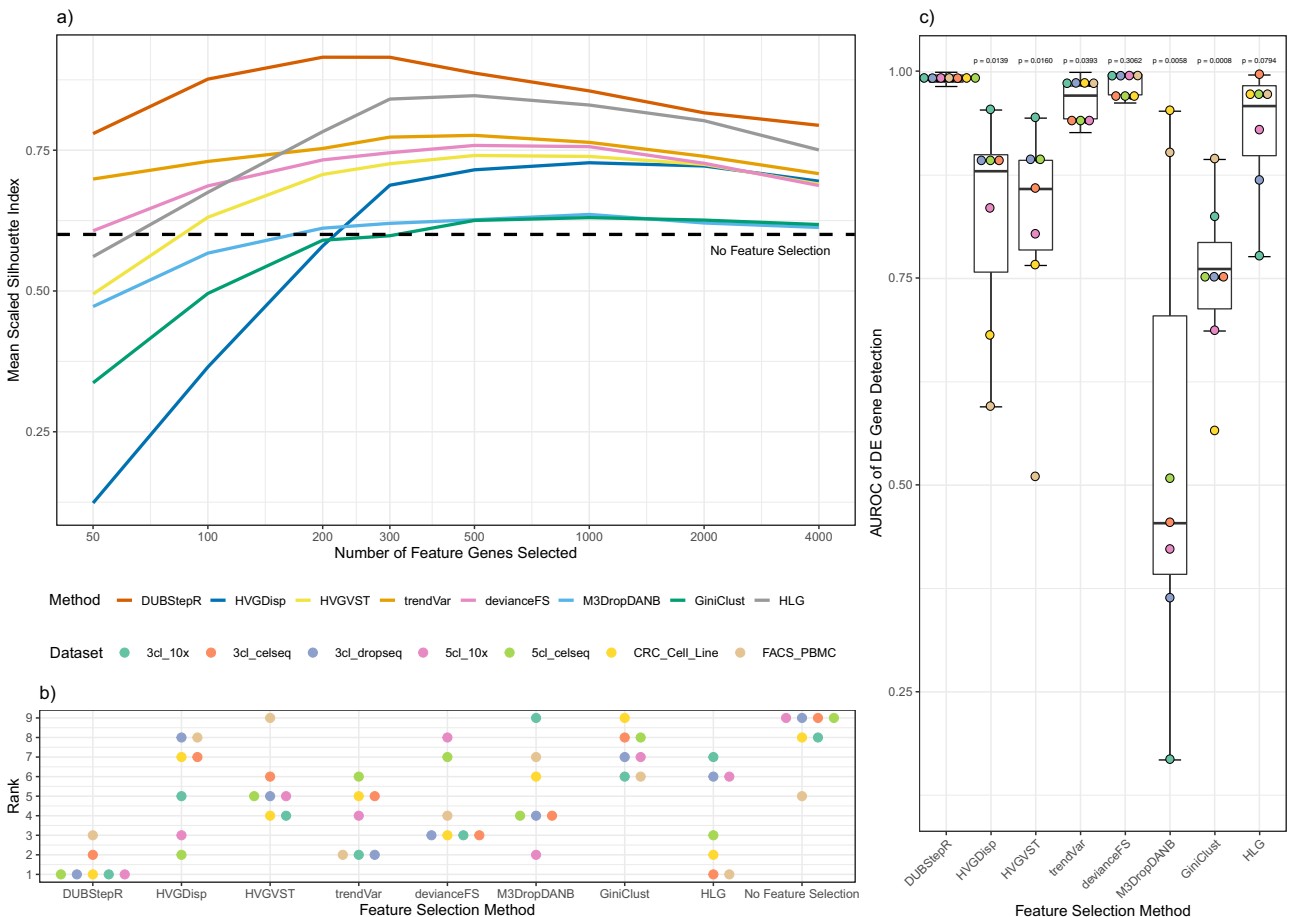

**Fig. 4 Benchmarking feature selection methods. a** Mean scaled silhouette index of feature sets ranging from 50 to 4000 features. **b** Rank distribution of feature selection methods. For each dataset, the methods are ranked from 1 to 9 by their best SI across all feature set sizes. **c** AUROC of DE gene detection. For all boxes, the middle line represents the median, the lower and upper box limits correspond to the first and third quartiles, the upper and lower whiskers extend upto 1.5*IQR from the top and bottom of the box, respectively (where IQR is the inter-quartile range). Points beyond the whiskers are outliers and are plotted individually. All p-values were calculated using two-sided Student's T-tests ($n = 7$). Refer to Supp. Table S6 for details regarding the datasets.

neighbors should be minimal relative to the distance between random pairs of cells, and thus DI should be maximal (Fig. 5a, b). Empirically, we found that DI and SI were indeed positively correlated and tended to reach their maxima at approximately the same feature set size (Fig. 5c). Further, for 5 out of the 7 benchmarking datasets, the feature set with the highest DI also maximized SI (Fig. 5d). Since our earlier analysis only tested a discrete number of feature set sizes (Fig. 4a; Supp. Note 3A, B), the DI-guided approach even improved on the maximum SI in 2 cases (Fig. 5d). One additional advantage of the DI is that it is relatively straightforward to compute, since the numerator is proportional to the square of the Frobenius norm of the gene expression matrix ("Methods"). By default, DUBStepR therefore selects the feature set size that maximizes DI.

**DUBStepR robustly detects rare cell types and cryptic cell states in rheumatoid arthritis samples.** The above quantitative benchmarking analyses were largely based on detection of common cell types (>10% of all cells) in cell lines or FACS-purified cell populations from healthy donors. To demonstrate the ability of DUBStepR to cluster cells from a complex primary sample, we generated scRNA-seq data from 8312 PBMCs from four rheumatoid arthritis (RA) patients ("Methods"). In this case, since the "true" cell type labels were unknown, our objective was to qualitatively compare results from the various feature selection

methods. We used SingleR[21] to select the T and NK cell subset (5329 cells; "Methods") since this cell population is challenging to sub-cluster by conventional methods, despite its relevance to inflammatory phenotypes. DUBStepR (with DI optimization) identified 10 discrete subtypes in this dataset, with sharply distinct gene expression signatures (Fig. 6a; Supp. Fig. S5). These included four rare cell clusters that were undetected or only partially detected by the other feature selection methods: red blood cells (RBCs, 1.8%), proliferating cells (2%), platelet-T doublets (3.4%), and platelet-NK doublets (3%) (Fig. 6b; Supp. Figs. S5, S6). While RBCs reflect contamination during PBMC isolation, platelet-lymphocyte complexes and proliferating T cells regulated by *KIAA0101* are thought to play a role in the pathophysiology of RA[22–24] (Supp. Fig. S5).

In addition to detecting multiple rare cell types, DUBStepR identified a dichotomy in CD4+ T, CD8+ T, and NK cells, defined by coordinated differential expression of *SET, C1orf56, C16orf54, CDC42SE1,* and *HNRNPH1* (Supp. Fig. S5), all of which have been previously identified as markers of a latently infected T cell subtype in HIV[25]. Once again, DUBStepR was the only feature selection method to clearly distinguish these cell states (Supp. Figs. S5, S6). In summary, DUBStepR was the only feature selection algorithm that robustly detected common and rare cell types and subtypes in this complex primary lymphocyte population.

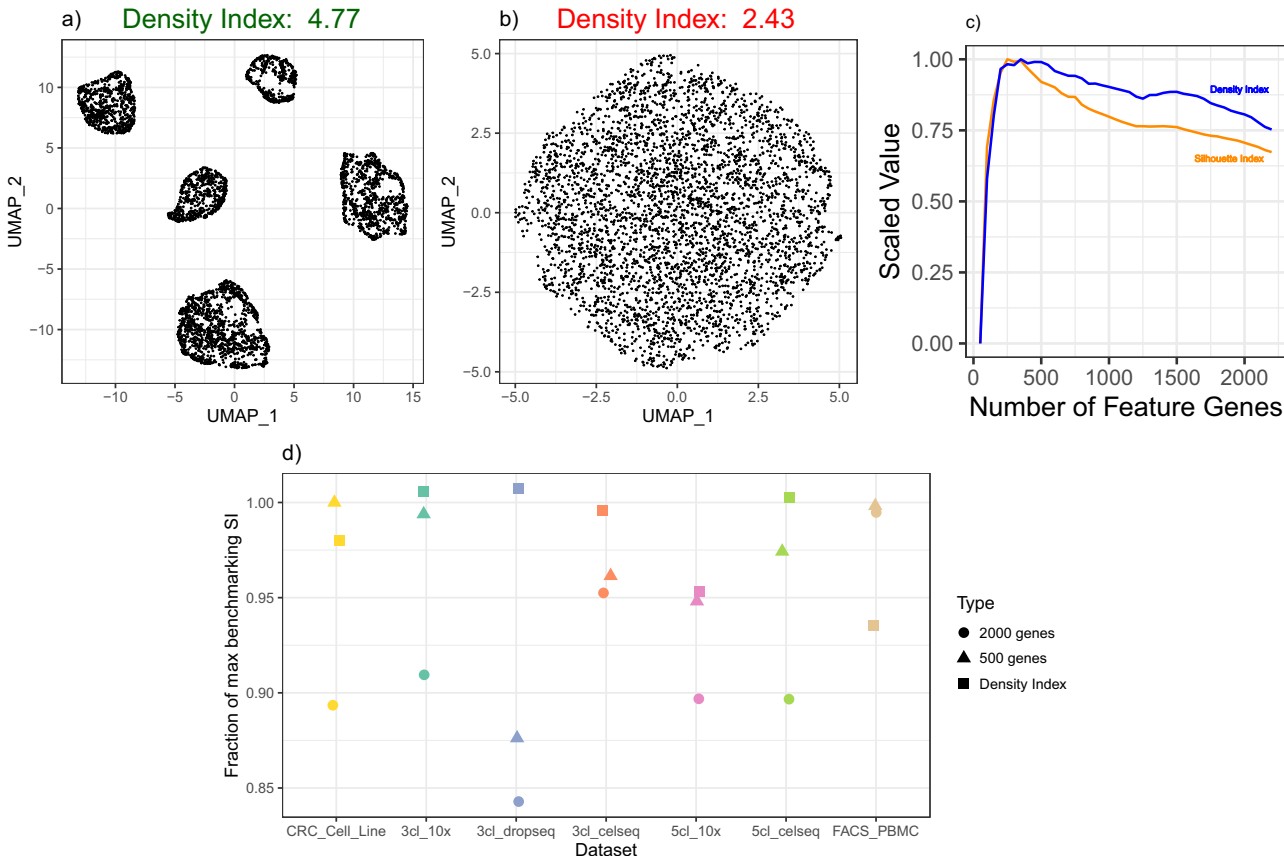

**Fig. 5 The density index. a**, **b** UMAP visualizations of the 5 cell lines comparing the local neighborhood density of feature spaces in (**a**) good feature selection versus (**b**) poor feature selection. **c** Comparison of optimal density index and silhouette index over feature set size range. **d** Fraction of the maximum benchmarking SI achieved using feature set sizes determined by the density index as compared to fixing feature set sizes at 500 genes and 2000 genes (refer to Supp. Table S6 for details regarding the datasets).

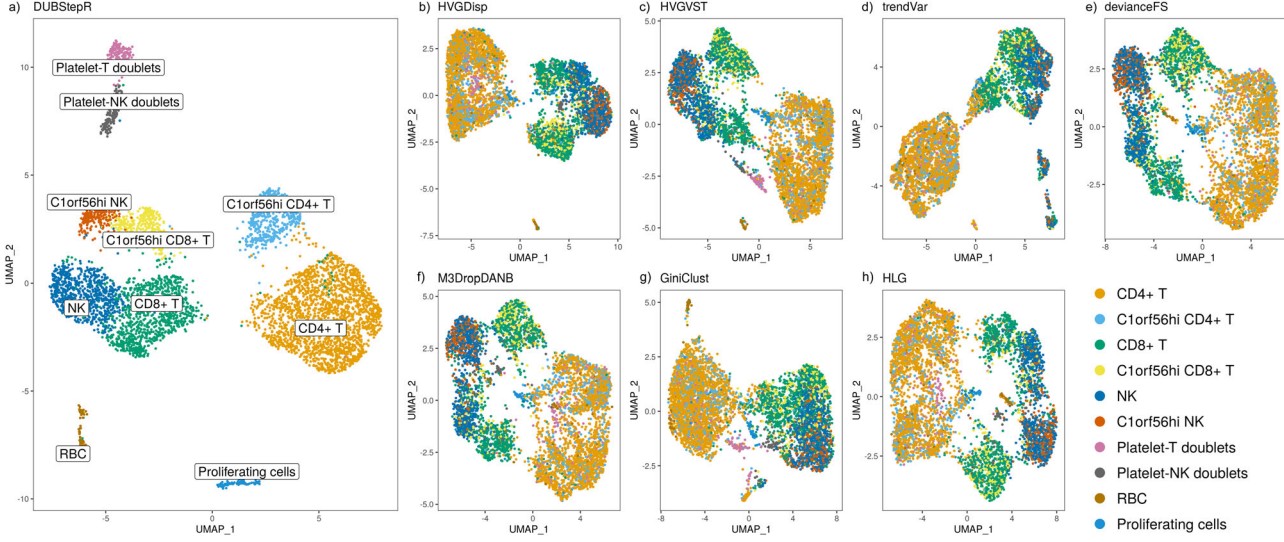

**Fig. 6 Analysis of lymphocyte population from rheumatoid arthritis patient PBMCs. a** UMAP visualization of clusters identified by DUBStepR. **b–h** UMAP visualization of cell clusters identified by DUBStepR using features selected by the 7 other feature selection methods; **b** HVGDisp, **c** HVGVST, **d** trendVar, **e** devianceFS, **f** M3DropDANB, **g** GiniClust, and **h** HLG.

**DUBStepR generalizes to scATAC-seq data**. Feature selection is typically not performed on scATAC-seq data, since their almost binary nature (most genomic bins have zero or one count) renders them refractory to conventional single-cell feature selection techniques based on variance-mean comparison[26]. However, since the logic of feature correlations applies even to binary or almost binary data, we hypothesized that DUBStepR could also improve the quality of cell type inferences from this data type. To

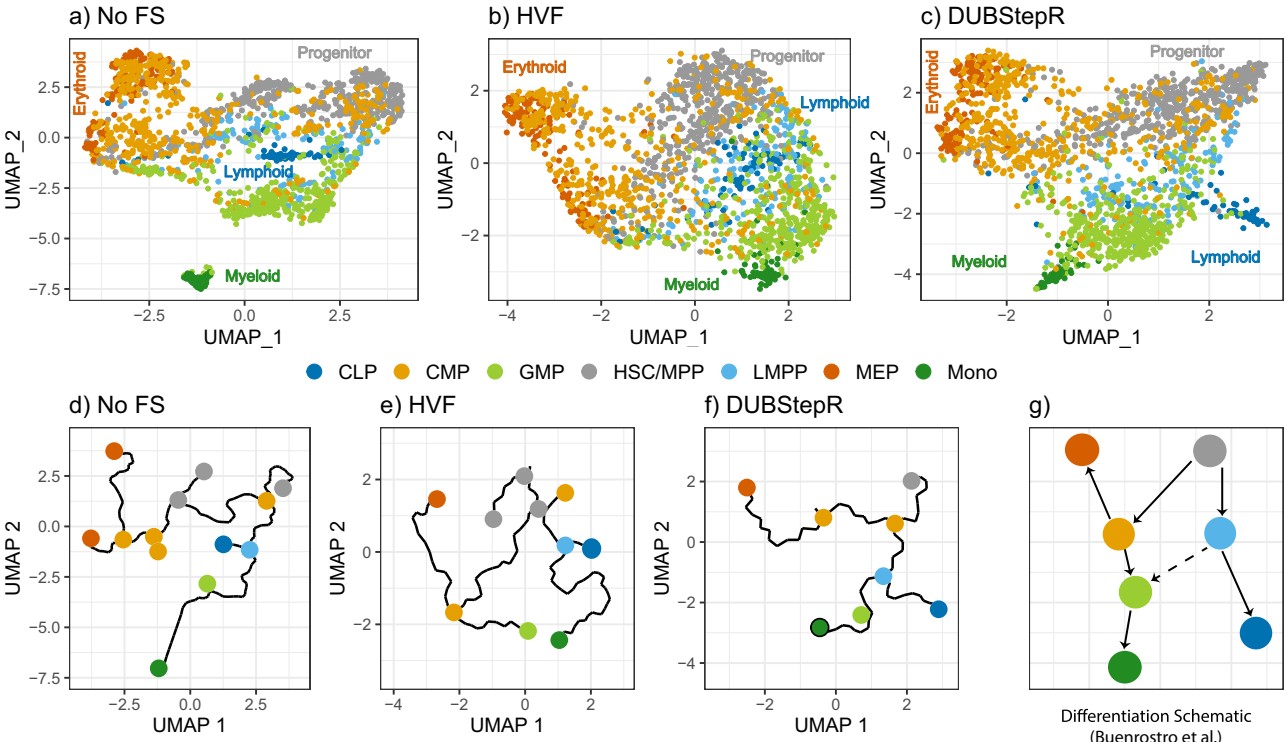

**Fig. 7 Comparison of feature selection on scATAC-seq data from Buenrostro et al.[27].** **a–c** UMAP visualizations of cells colored by FACS-sorted cell type labels, with trajectories constructed using Monocle 3. **a** All peaks (no feature selection), **b** peaks selected by highly variable features (HVF), **c** peaks selected by DUBStepR. **d–f** Monocle-generated trajectories on the corresponding UMAP visualizations. **g** Hematopoietic differentiation schematic. CLP common lymphoid progenitor, CMP common myeloid progenitor, GMP granulocyte/macrophage progenitor, HSC hematopoietic stem cell, LMPP lymphoid-primed multipotent progenitor, MEP megakaryocyte/erythroid progenitor, Mono monocyte, MPP multipotent progenitor.

test this hypothesis, we applied DUBStepR to scATAC-seq data from eight FACS-purified subpopulations of human bone marrow cells[27] (Supp. Note 2D). In contrast to the common approach of using all scATAC-seq peaks, we found that peaks selected by DUBStepR more clearly revealed the emergence of the three major lineages from hematopoietic stem cells: lymphoid, myeloid, and megakarocyte/erythroid (Fig. 7). Specifically, trajectory analysis using `Monocle 3`[20] yielded a topology that matched the known hematopoietic differentiation hierarchy[27] (Fig. 7g) only in the case of DUBStepR (Fig. 7d–f).

## Discussion

DUBStepR is based on the intuition that cell-type-specific marker genes tend to be well correlated with each other, i.e., they typically have strong positive and negative correlations with other marker genes. After filtering genes based on a correlation range score, DUBStepR exploits structure in the gene–gene correlation matrix to prioritize genes as features for clustering. To benchmark this feature selection strategy, we used a stringently defined collection of single-cell datasets for which cell type annotations could be independently ascertained[28]. Note that this avoids the circularity of defining the ground truth based on the output of one of the algorithms being tested. Results from our benchmarking analyses indicate that, regardless of feature set size, DUBStepR separates cell types more clearly other methods (Fig. 4a, b). This observation is further corroborated by the fact that DUBStepR predicts cell-type-specific marker genes substantially more accurately than other methods (Fig. 4c). Thus, our results demonstrate that gene–gene correlations, which are ignored by conventional feature selection algorithms, provide a powerful basis for feature selection.

The plummeting cost of sequencing, coupled with rapid progress in single-cell technologies, has made scalability an essential feature of novel single-cell algorithms. DUBStepR scales effectively to datasets of over a million cells without sharp increases in time or memory consumption (Supp. Fig. S4). Thus, the method is likely to scale well beyond a million cells. A major contributor to the algorithm's scalability is the fact that, once the gene–gene correlation matrix is constructed, the time and memory complexity of downstream steps is constant with respect to the number of cells.

Intriguingly, DUBStepR approaches its maximum silhouette index value at 200–500 feature genes (Supp. Fig. S3), which is well below the default feature set size of 2000 used in most single-cell studies[10,12]. Thus, our results suggest that, if feature selection is optimized, it may not be necessary to select a larger number of feature genes. Note, however, that the optimum feature set size can vary across datasets (Supp. Fig. S3). Selecting a fixed number of feature genes for all datasets could therefore result in sub-optimal clustering (Fig. 5d).

From the perspective of cell clustering, the optimal feature set size is that which maximizes cell type separation in feature space, which can be quantified using the SI. As an indirect correlate of cell type separation, we have defined a measure of the inhomogeneity or "clumpiness" of cells in feature space, which we term the density index (DI). To our knowledge, DI is the only metric for scoring feature gene sets based on the distribution of cells in feature space. Our results suggest that the DI correlates with the SI, and that cluster separation is improved in most cases when the feature set is chosen to maximize DI. Another important advantage of the DI is that it is computationally straightforward to calculate from the Frobenius norm of the data matrix. It is possible that the DI measure could also be

applied to other stages of the clustering pipeline, including dimensionality reduction (selecting the optimal number of PCs) and evaluation of normalization strategies.

Interestingly, although DUBStepR was not specifically designed to detect rare cell types, it nevertheless substantially outperformed all other methods at detecting multiple cell populations present at low frequency (<4%) in a complex primary PBMC sample (Fig. 6). Notably, the rare populations identified by DUBStepR-included RBCs and platelet-containing doublets, which should not have been present in the T/NK population. It is likely that these cells were mis-classified as T/NK by `SingleR` due to the absence of platelet and RBC transcriptomes in the reference panel. In addition, DUBStepR greatly outperformed all other feature selection methods at detecting T and NK cell sub-populations over a range of higher frequencies (4.9−9.5%) in the same dataset. Given that this dataset posed the greatest challenge in terms of clustering difficulty, it is remarkable that DUBStepR provided a major, qualitative improvement over all other feature selection methods.

Algorithmic pipelines for single-cell epigenomic data, for example scATAC-seq, typically do not incorporate a formal feature selection step[26,29]. In most cases, such pipelines merely discard genomic bins at the extremes of high and low sequence coverage. This is because the sparsity and near-binary nature of single-cell epigenomic data reduces the efficacy of conventional feature selection based on mean-variance analysis. Since DUBStepR uses an orthogonal strategy based on correlations between features, it is less vulnerable to the limitations of single-cell epigenomics data (Fig. 7). Thus, DUBStepR opens up the possibility of incorporating a feature selection step in single-cell epigenomic pipelines, including scATAC-seq, scChIP-seq, and single-cell methylome sequencing.

## Methods
### DUBStepR methodology
*Gene filtering.* By default, DUBStepR filters out genes that are not expressed in at least 5% of cells. We allow the user to adjust this parameter if they are interested in genes that are more sparsely expressed in their dataset. In addition, mitochondrial genes, ribosomal genes, and pseudogenes are identified using gene symbols or Ensembl IDs for human, mouse, and rat datasets using the latest Ensembl references downloaded from BioMart[30].

*Correlation range.* Correlation range $c_i$ for gene $i$ can be defined for a gene–gene correlation matrix $G$ as

$$c_i = \max_3(G_i) - 0.75 \cdot \min(G_i),$$ (1)

where $\max_h(G_i)$ refers to the $h$th-largest correlation value in column $i$ of $G$. Correlation range uses the second-largest non-self correlation value (3rd largest value in GGC column) to calculate the range, so as to protect against genes with over-lapping 5′ or 3′ exons[31].

The minimum correlation value has been down-weighted to 0.75 to give greater importance to stronger positive correlations over negative correlations.

We first binned genes based on their mean expression level, as mean expression tends to correlate with technical noise[32]. In each bin, we compute a z-score of the correlation range of gene $i$ as

$$z_i = \frac{c_i - \mu_c}{\sigma_c},$$ (2)

where $\mu_c$ is the mean correlation range of the gene and $\sigma_c$ refers to variance in the correlation range scores of a gene. Genes with a z-score $\leq 0.7$ are filtered out at this step.

*Stepwise regression.* We define the stepwise regression equation as

$$G = g\mathbf{w}^T + \epsilon,$$ (3)

where $G$ is the column-wise zero-centered gene–gene correlation matrix, $\mathbf{g}$ is the column of the matrix $G$ to be regressed out, $\epsilon$ is the matrix of residuals and $\mathbf{w}$ is a vector of regression coefficients. The squared error ($\epsilon^T\epsilon$) is minimized when

$$\mathbf{w}^T = \frac{\mathbf{g}^T G}{\mathbf{g}^T \mathbf{g}}$$ (4)

Thus,

$$G = \frac{\mathbf{g}\mathbf{g}^T G}{\mathbf{g}^T \mathbf{g}} + \epsilon.$$ (5)

We calculate variance explained by the regression step as $V = \| G - \epsilon \|_F^2$, where $F$ indicates the Frobenius norm. To efficiently compute $V$ for all genes, we define $X = G^T G$ and $\mathbf{x}$ as the row of $X$ corresponding to gene $\mathbf{g}$. Thus, $\mathbf{x} = \mathbf{g}^T G$. We can simplify $V$ as

$$\begin{aligned} V &= \| G - \epsilon \|_F^2 \\ &= \left\| \frac{\mathbf{g}\mathbf{g}^T G}{(\mathbf{g}^T \mathbf{g})} \right\|_F^2 \\ &= \frac{\| \mathbf{g}\mathbf{x} \|_F^2}{(\mathbf{g}^T \mathbf{g})^2} \\ &= \frac{\text{Tr}((\mathbf{g}\mathbf{x})^T (\mathbf{g}\mathbf{x}))}{(\mathbf{g}^T \mathbf{g})^2} \\ &= \frac{\text{Tr}(\mathbf{x}^T \mathbf{x})}{\mathbf{g}^T \mathbf{g}} \\ &= \frac{\mathbf{x}\mathbf{x}^T}{\mathbf{g}^T \mathbf{g}}. \end{aligned}$$ (6)

Thus, we can use a single matrix multiplication $G^T G$ to efficiently calculate variance explained by each gene in the gene–gene correlation matrix, and then regress out the gene explaining the greatest variance. The residual from each step $k$ is then used as the gene–gene correlation matrix for the next step. In other words,

$$\begin{aligned} G_k &= \mathbf{g}_k \mathbf{w}_k^T + \epsilon_k \\ G_{k+1} &= \epsilon_k. \end{aligned}$$ (7)

For computational efficiency, we repeat this regression step 30 times and then assume that the next 70 steps explain the amount of variance as the 30th step, giving a total of 100 steps. We observed that this shorter procedure had little or no impact on the results, since the variance explained changed only marginally beyond the 30th step.

To select the genes contributing to the major directions in $G$, we use the elbow point on a scree plot. Elbow point computation is described in Supp. Note 1B. The genes that are regressed out upto the elbow point form the "seed" gene set.

*Guilt-by-association.* Guilt-by-association, also known as label propagation through a network, allows DUBStepR to determine a robust ordering of features in an iterative manner. Once the seed genes have been determined, the gene with the strongest Pearson correlation to any of the seed genes is first identified. This gene is then added to the seed genes, thereby expanding the feature set. This feature set (now consisting of the seed genes and the newly added feature gene) is then used to identify the next most strongly correlated gene, which is again added to the feature set. By iteratively repeating the latter step, DUBStepR propagates through the gene–gene correlation network until the feature set has reached its final size.

We have developed a custom implementation of this guilt-by-association approach as part of the DUBStepR package in R, the source code for which is available on our GitHub repository (see "Code availability" section).

*Density index.* For a given feature set, PCA[5] is performed on the gene expression matrix and the top $D$ principal components (PCs) are selected, where $D$ is a user-specified parameter with a default value of 20. Let $M$ be the matrix of embeddings of the gene expression vectors of $N$ cells in $D$ principal components. The root-mean-squared distance $d_{rms}$ between pairs of cells $i$ and $j$ can be calculated as

$$d_{rms} = \sqrt{<d_{i,j}^2>} = \sqrt{<\Sigma_{p=1}^D (M_{i,p} - M_{j,p})^2>},$$ (8)

where $<>$ denotes the average over all pairs $\{(i,j)|i\in[1,N], j\in[1,N]\}$. Note that, for simplicity of the final result, we include pairs in which $i = j$. This can be further simplified as follows:

$$\begin{aligned} d_{rms} &= \sqrt{<\Sigma_{p=1}^D (M_{i,p}^2 + M_{j,p}^2 - 2M_{i,p}M_{j,p})>} \\ &= \sqrt{\Sigma_{p=1}^D (<M_{i,p}^2> + <M_{j,p}^2> - 2<M_{i,p}M_{j,p}>)} \\ &= \sqrt{\Sigma_{p=1}^D (<M_{i,p}^2> + <M_{j,p}^2>)} \\ &= \sqrt{2\Sigma_{p=1}^D <M_{i,p}^2>} \\ &= \sqrt{\frac{2}{N}} \| M \|_F. \end{aligned}$$ (9)

In the above derivation, the mean product term $<M_{i,p}M_{j,p}>$ is zero because $M_{i,p}$ and $M_{j,p}$ have zero mean across $i$ and $j$, respectively. Let $k_i$ denote the average

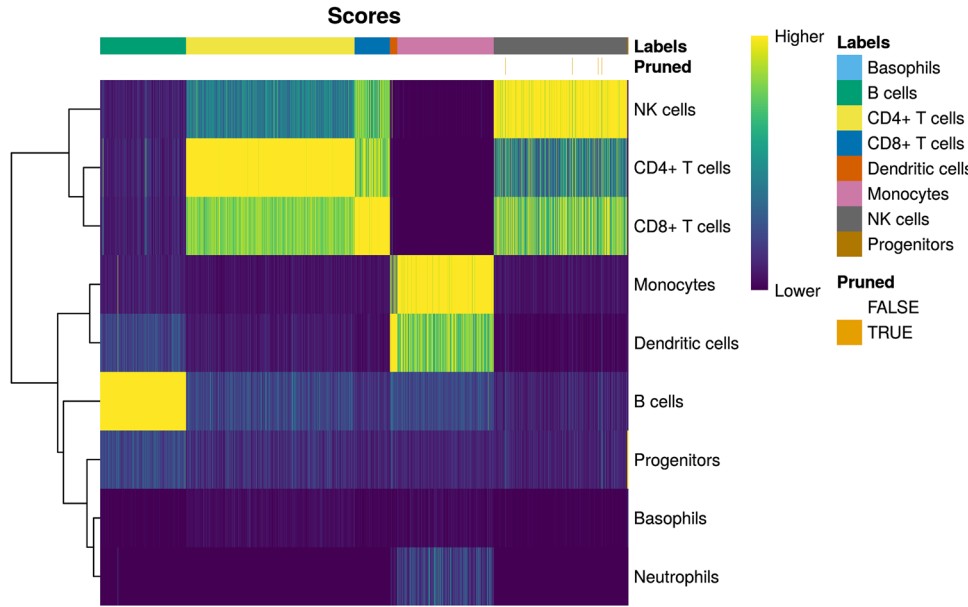

**Fig. 8 Heatmap showing** `SingleR` **scores for cells in the RA PBMC dataset computed against the Monaco immune reference**[36]**.** Higher score indicates greater likelihood of the cell being of that corresponding cell type. Cells pruned by `SingleR` are indicated in orange.

**Table 2 Number of cells of each cell type annotated by** `SingleR` **that were obtained from each patient in the RA PBMC dataset.**

| Cell type | Patient RA_1 | Patient RA_2 | Patient RA_3 | Patient RA_4 |
|---|---|---|---|---|
| Basophils | 1 | 0 | 0 | 0 |
| B cells | 288 | 550 | 112 | 404 |
| CD4+ T cells | 492 | 1142 | 501 | 523 |
| CD8+ T cells | 189 | 48 | 242 | 78 |
| Dendritic cells | 20 | 14 | 43 | 40 |
| Monocytes | 351 | 143 | 535 | 492 |
| NK cells | 710 | 194 | 680 | 500 |
| Progenitors | 2 | 5 | 6 | 7 |
| Total | 2053 | 2096 | 2119 | 2044 |

distance of cell $i$ from its $k$ nearest neighbors, and $k_m$ denote the mean of $k_i$ across all cells. We define the DI as

$$\text{DI} = \frac{d_{rms}}{k_m} = \sqrt{\frac{2}{N}} \frac{\| M \|_F}{k_m}. \qquad (10)$$

**Rheumatoid arthritis dataset**

*Patient sample collection.* Fresh blood samples of patients diagnosed with rheumatoid arthritis were collected at Tan Tock Seng Hospital, Department of Rheumatology, Allergy & Immunology, Singapore, and were transferred to Genome Institute of Singapore for further processing. All technical procedures and protocols for the recruitment, blood collection, and PBMCs isolation were reviewed and approved by the Institutional Review Board (IRB) at the National Healthcare Group Domain Specific Review Board (NHG DSRB), Singapore (Reg. no. 2016/00899).

*Single-cell RNA sequencing.* For each sample, fresh PBMCs were isolated from 5 ml Sodium Heparin tubes using standard Ficoll-Hypaque density gradient centrifugation[33]. Briefly, blood samples were diluted by an equal volume of Phosphate-buffered saline (PBS) containing 2% fetal bovine serum (FBS) (STEMCELL, catalog #07905) and were gently added on top of Lymphopreb™ density gradient medium (STEMCELL, catalog #07801) in the SepMate™ (STEMCELL, catalog #15420) tubes. The tubes were centrifuged at $1200 \times g$ for 15 min and the upper layer containing the enriched PBMCs and plasma was collected into a new falcon tube to wash with PBS + 2% FBS and centrifuge at $300 \times g$ for 8 min. After counting the cells, isolated PBMCs were divided into two or three vials and frozen down in FBS containing 10% dimethylsulfoxide (DMSO) for downstream experiments. Immediately after isolation, fresh PBMCs were washed, filtered (using cell strainer), and re-suspended in PBS containing 0.04 bovine serum albumin. Single-cell suspensions from four samples were mixed at the final concentration of $10^6$ cells/ml and the mixed suspension was loaded into a 10x Chromium Instrument to target total number of 3000 cells from each patient ($n = 4$).

GEM-RT reactions, cDNA synthesis, and library preparations were performed using Single Cell 3′ v2 10x Reagent™ Kit according to manufacturer protocols. The single-cell libraries were run onto the Illumina HiSeq® 4000 platform as prescribed by 10x Genomics. The raw base call (BCL) files from the sequencer were processed through 10x Genomics Cell Ranger 2.1.1 analysis pipelines. First, the `mkfastq` pipeline was run to generate FASTQ files, followed by read alignment to the *hg19* genome reference using the `count` pipeline. The raw counts matrix generated by the `count` pipeline was loaded into R for further analysis.

*Demultiplexing pooled samples.* To facilitate demultiplexing, the patients were genotyped using the Illumina Infinium® HTS Assay following the manufacturer's protocol. Reads from the sequencer were attributed to each of the four patients using `Demuxlet`[34] at default settings. For each library, the corresponding genotyping data (VCF file) and scRNA-seq data (BAM file) were imported into `Demuxlet` in order to infer the sample source for each cell barcode.

*Data preprocessing.* The raw data files were used to generate a `Seurat` object, including only features that were detected in a minimum of 3 cells and cells having at least 200 uniquely detected features. Additional filtering was performed to remove cells having >10% mitochondrial rate (calculated using the PercentageFeatureSet function in `Seurat`[35]). Further, only cells with unique feature counts between 200 and 2500 were retained, resulting in 8350 cells. The data was then log-normalized using the NormalizeData function in `Seurat`[35] at default settings.

*Isolating T & NK cells.* The normalized data were then annotated using `SingleR`[21], with the Monaco et al. immune dataset[36] as the reference. The SingleR function was run at default settings, using the log-normalized counts as input. Thirty-eight cells were pruned by `SingleR`, leaving a total of 8312 annotated cells. Of these, cells annotated as CD4+ T cells, CD8+ T cells, or NK cells were isolated (Fig. 8; Table 2).

*DUBStepR analysis*. Due to the lower expression of genes in the T and NK cell population, we modified DUBStepR's gene filtering threshold to filter out genes expressed in less than 1% of the cells. The `Seurat` package was used for downstream processing. First, the feature genes were zero-centered and scaled, and PCA was performed. The top 12 PCs were selected using the elbow plot of variance explained by the PCs. Clustering was performed using the Louvain graph-based approach—using the FindNeighbors function with 12 PCs and FindClusters at default parameter settings. UMAP coordinates were computed using the cell embeddings in 12 PCs.

*Other feature selection methods*. All other feature selection methods (HVGDisp, HVGVST, trendVar, devianceFS, M3DropDANB, GiniClust, and HLG) were run at default settings. To showcase the result of HVGVST, we used the `Seurat` package to cluster cells, as described above. Similar to the DUBStepR result, 12 PCs were used for both clustering and UMAP visualization.

**Reporting summary**. Further information on research design is available in the Nature Research Reporting Summary linked to this article.

## Data availability

Raw data for the 3cl_10x, 3cl_dropseq, 3cl_celseq, 5cl_10x, and 5cl_celseq datasets are available under GEO SuperSeries GSE118767. The CRC Cell Line dataset is deposited in GEO under the accession code GSE81861. The FACS PBMC dataset is freely available on the 10x Genomics website (https://support.10xgenomics.com/single-cell-gene-expression/datasets). The Mouse Organogenesis Cell Atlas dataset used to benchmark computational scalability is freely available in the Seattle Organismal Molecular Atlases (SOMA) Data Portal (https://oncoscape.v3.sttrcancer.org/atlas.gs.washington.edu.mouse.rna/downloads). Finally, the accession number for the single-cell ATAC sequencing data reported in this paper is GEO: [GSE96772]. Processed data used for generating the figures in this paper, including our in-house-generated RA PBMC scRNA-seq data, are available on Zenodo at https://doi.org/10.5281/zenodo.4072260. The raw FASTQ files for the in-house-generated rheumatoid arthritis PBMC data are part of an ongoing large-scale single-cell project which requires controlled access. Access requests should be directed to Shyam Prabhakar (prabhakars@gis.a-star.edu.sg) and Leong Khai Pang (khai_pang_leong@ttsh.com.sg), and will be responded to within 3 working days.

## Code availability

DUBStepR is available as an R package on CRAN (https://CRAN.R-project.org/package=DUBStepR), and is well documented for easy integration into the `Seurat` pipeline. The source code is freely available on GitHub (https://github.com/prabhakarlab/DUBStepR)[37]. R scripts for generating all the figures in this paper are available on Zenodo at https://doi.org/10.5281/zenodo.4072260[38].

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

## Acknowledgements

The authors would like to acknowledge Nancy Q. Zhao for her insights on the rheumatoid arthritis dataset analysis, Lavanya M. Iyer for her mentorship and guidance, and all other members of the Prabhakar lab for critical feedback and discussions. This publication is part of the Human Cell Atlas - https://www.humancellatlas.org/publications. The work was supported by the Agency for Science, Technology and Research, Singapore [CDAP201703-172-76-00056, H17/01/a0/007, and IAF-PP-H18/01/a0/020], National Medical Research Council Centre grants (CG12Aug17 and CGAug16M012), and funding from the Department of Rheumatology, Allergy and Immunology, Tan Tock Seng Hospital.

## Author contributions

B.R., W.S., and S.P. designed the DUBStepR algorithm, with critical inputs from J.P., V.S., F.S., and I.J. B.R. and W.S. developed the software and performed benchmarking analyses with assistance from K.M., J.P., R.X., F.A., F.S., N.A.R., and M.G.L.L. F.S. also provided critical inputs relating to the scATAC-seq analysis. M.A.H., J.M.Y.Y., E.T.K., and L.K.P. generated the human RA PBMC scRNA-seq dataset. B.R. and S.P. wrote the manuscript.

## Competing interests

The authors declare no competing interests.
