## [Peer Review File · Nature Communications]

DUBStepR is a scalable correlation-based feature selection method for accurately clustering single-cell dataReviewers' Comments:

Reviewer #1:

Remarks to the Author:

In this paper, the authors present a new method for feature/gene selection in scRNA-seq data. This method is mainly based on gene-gene correlations, and is able to select features/genes that lead to improved performance in subsequent clustering analysis.

My major concerns about this paper are as follows:

1. According to the definition of correlation range score in the Method section, the max possible value for this score is 1.75. However, the boxplot in figure 1d showed scores exceeding 2. This does not seem to be correct.

2. The definition of range score seems to be problematic. Given this definition of $c_i = \max_3(G_i) - 0.75\min(G_i)$. Think about two hypothetical cases:

(1) If gene i's correlation with all other genes are all 0.5, the range score will be $0.5 - 0.75 \times 0.5 = 0.125$.

(2) If gene i's correlation with all other genes are all -0.5, the range score will be $-0.5 - 0.75 \times (-0.5) = -0.125$

First of all, it would be ideal that the range score for these two hypothetical cases are both 0. Second, even if not, the range score for these two hypothetical cases should at least be the same. More such make-up extreme examples can be made to show that such a definition can be problematic.

3. Back to Figure 1D, it shows that the smallest correlation range value can be lower than -1. I really cannot imagine an example that will lead to a -1 score, according to the range score definition.

4. In the stepwise regression procedure, the choice of a total of 100 steps is not well justified. In method section, there is an assumption that the first 30 steps and the next 70 steps explain the same amount of variance. This is likely dataset-dependent. Since the total number step affects the identification of the elbow, the choices made here need to be properly evaluated and justified.

5. With respect to cell clustering, the 7 scRNA-seq datasets benchmarked here are all relative easy datasets. 6 of the 7 are cells from multiple cell lines, which are easy to cluster. The last one of the 7 contains cells from FACS sorted subset of PBMCs, which is also easy to cluster because confusing/challenging cells in-between well established cell types/states are not going to be in the FACS gates. Therefore, the comparisons of algorithms are all done on easy datasets. Analysis based on datasets with "normal" level of difficult will make the comparison more convincing.

6. The first paragraph in page 5, the statement of "For optimal cell type clustering, a feature selection algorithm should ideally select only cell-type-specific genes (DE genes) as features", and the evaluation figure 3c based on this statement, both are flawed. Yes, cell-type-specific genes (for the known cell types / cell lines) should be selected so that these cell types/lines can be separated. However, selecting other genes does not mean that the selection algorithm is bad. If one cell line has a lot of heterogeneity (several subtypes), a good feature selection algorithm should select genes that can help cluster the subtypes within that cell line, but these genes are not "cell-type-specific genes" in this evaluation. Therefore, the proposed method winning this evaluation shown in Figure 3c does not really provide a convincing argument that the proposed method is better. It is actually misleading.

7. The density index idea can also be problematic. If there exist a set of genes that leads to many many cell clusters and each cluster has more than k cells, the DI score will be good. However, such a set of genes certainly are not good features for clustering, and the SI score based on the known cell clusters is going to be poor. This is one counter example showing that the DI score may not be a good approximation of the SI score in certain scenarios. It is good that Figure 4c shows the correlation between SI and DI in the cell line data. But again, the datasets analyzed here are all easy datasets for

cell clustering, limiting the generality of the conclusions.

8. The analysis/comparison on scATACseq data in Figure 7 is not sufficiently described. Looking at Figure 7, it is unclear what the figure shows the proposed feature selection is better than not doing feature selection at all. More discussions/interpretations are needed to make this section meaningful.

Overall, multiple aspects of this paper seem to be problematic to me. Therefore, I am afraid that I cannot give a positive recommendation for this paper.

Reviewer #2:

Remarks to the Author:

The authors describe a novel and effective approach for the feature selection method in scRNA-seq data to improve single cell clustering. This is an important problem in the field that could be widely applied for aiding insights from sparse complex multidimensional data.

DUBSTepR first carries out feature selection based on the idea that top differential genes of same cell types correlate highly with each other, to generate a gene-gene correlation matrix. An optimal number of feature genes is then determined, which is very important since if it is too high, the separation of cells would not be optimal and if it is too few, it may not work well due to the sparse nature of the data, both leading to sub-optimal clustering. Often researchers need to try different number of genes to use (highly variable genes being one of the most common method currently used) and decide based on the outcome, which tends to be subjective. Thus, the approach DUBSTepR takes to determine the optimal number of feature genes is highly appreciated.

The authors show that DUBStepR outperforms compared to other existing methods by multiple measures. Excitingly, although DUBStepR was not originally developed to detect rare cell types, they have shown it is also promising in this regard. Furthermore, DUBStepR appears to perform well with scATAC-seq datasets and can in principle be applied to other single-cell epigenomic datasets such as scChIP-seq and DNA methylation datasets. The authors should be praised for providing all code and data in publicly available repositories and implementing the tool as a (creatively named) R package to facilitate wider use.

Overall it is a very well-written paper with some interesting mathematical insights. DUBStepR is worth bringing attention of to the community of single cell genomics and recommend it to be published upon taking in the points below.

Major comments:

I would like to see how much better the DUBStepR performs to other methods on a more complex sample that contains many more cell types than those that have been used in the manuscript, such as mouse lung. Mouse lung would also be a good tissue to test whether the DUBStepR would detect rare cell types, as claimed by the authors, such as ILC2s. ILC2s are present in the lung but not detected with currently available data processing tools due to its low number in steady state. The single cell genomics community would appreciate to see this.

Some of the single-cell techniques demonstrated (10x Genomics, Drop-seq) use UMI counts whereas others (Smart-Seq2) use read counts. Does feature selection work better with UMI or can it be applied to either dataset?

Several claims are made regarding scATAC-seq data. How does step-wise regression on a gene-gene correlation matrix work differently on a binary scATAC or scChIP peak count matrix? Would you expect it to work similarly on Beta-values for single-cell bisulfite sequencing for DNA methylation? More supporting data to back up the claims about epigenomics applications would be helpful.

For the scATAC-seq dataset, I am curious what peaks are feature selected by DUBStepR. Are they mostly peaks associated with coding or non-coding genes? Promoters or distal? Since enhancers are known to be more cell type-specific, overview of such categories of the peaks that were selected by DUBStepR would be of interest to the reviewers.

Several claims of computational performance are made but DUBStepR appears to have similar or greater cpu-time and memory requirements to other techniques (Fig. S3). Is this acceptable given the higher accuracy from iterative step-wise regression?

Minor points:

How does DUBStepR remove mitochondrial and ribosomal genes? Is this based on gene ID or symbols? If it is hardcoded for the human data demonstrated, is it possible to apply to other species with different gene naming conventions?

If 30 steps are performed as an approximation of 100 steps, it seems unnecessary to mention that 100 steps are used. The number of iterations appears to have been reduced for performance. This could be stated more unequivocally.

It would be helpful to distinguish datasets in Fig 3b (by colors, shapes, or numeric text labels).

The R package is now available on CRAN and the manuscript should be updated to reflect this.

<https://github.com/prabhakarlab/DUBStepR>
<https://CRAN.R-project.org/package=DUBStepR>

Reviewer #3:

Remarks to the Author:

Clustering analysis is a key component in the computational pipeline of Single-cell RNA-seq and ATAC-seq to understand the cellular heterogeneity and extract cell type/state markers. One of the important prior steps before clustering is to determine a suitable top variable feature (gene) set that largely captures differential expressed genes across cell types. In this study, the authors present a new framework based on gene expression correlation to select an optimal set of features by an optimized stepwise correlation, guilt-by-association, and newly invented density index (DI). They benchmarked the framework across the different platforms to validate its significant improvement on SI and DI scores than other software. The R package is fully integrated into the classic Seurat workflow and might be promising to become a routine replacement of HVGST in the future. However, the manuscript still needs to be largely improved to make the method and the validation more clear and solid, especially for the real biological datasets in the case studies. Please see the following comments.

Major comments:

- 1) The max3 function in the method part is not clear. Please provide the mathematical formula to define it.
- 2) In a routine clustering or trajectory analysis, we usually select a subset of 1000~2000 genes. In Figure 3a, when selecting 2000 features, the SI increase of DUBStepR is minor in comparison to

Seurat and HLG methods, but the DE metrics in Figure 3c is quite significant, given that in Figure 1, the correlation between correlation range score and log fold change is highly significant, is there any mathematical link between the computation of correlation range score and the method of differential gene expression analysis?

3) Please consider adding a state-of-art method deviance score and Pearson residual from GLM-PCA study and associated scry R package for feature selection of UMI-based single-cell RNA-seq, and provide corresponding SI and DI metrics in Figure 3a-c.

4) In the section of Guilt-by-association expands the feature set, authors claim each individual signature is represented by 2-5 genes, but Fig.2i select the elbow point by 20 genes. What does the signature mean here? Is it a cell cluster signature? Furthermore, DUBStepR detects a handful of seed genes in backward stepwise regression and expands the seed genes by recycling correlated genes from the candidate feature set in a forward way, could the two steps combined together to select the feature set once by a certain regression statistic such as BIC?

5) The description of Guilt-by-association is too brief in the main text to understand and ignored in the Methods section. From the text and Figure 5, it is hard to tell how to expand the feature set on the genes UMAP, is it conducted by finding the nearest genes of seed genes? The authors need to describe more details about this step. DUBStepR orders genes first by Guilt-by-association order, then correlation range. What is the metric output by the Guilt-by-association step? Is the Guilt-by-association metric more informative than the stepwise regression and correlation range?

6) DUBStepR application in sorted PBMC dataset is not a good example, since Basophil cell population is constituted with only 12 cells and the SI is equal to Seurat HVGST, it is not superior to use DUBStepR to identify five additional cells than the routine Seurat workflow, authors may need to find a better real biological dataset to justify the clustering improvement of DUBStepR to identify rare cell type.

7) It is not clear that DUBStepR surpasses other methods in Figure 7. Are the dimensional reduction and clustering of single-cell ATAC-seq based on the k-mers, peak counts, or gene scores? What's the SI and DI comparison among feature selection of DUBStepR and Seurat and all features based on the FACS-sorting information? Correlation measurement might be changed in DUBStepR since scATAC-seq is a binary matrix. Authors may consider the Jaccard index used by Snapatac as similarity measurement.

8) Would the feature selection proposed by DUBStepR improve another important downstream analysis dimensionality reduction such as PCA, UMAP, and TSNE? Are the UMAPs and clusters shown in the manuscript using the same feature selected in the DUBStepR?

Minor comments:

1) More information about normalization and clustering approaches of single-cell RNA-seq and single-cell ATAC-seq based on the selected features should be provided.

2) How many cells per lung adenocarcinoma cell line? Fig 1a cell type should be replaced by cell line.

3) Are other methods that failed to scale up due to software failure or consume endless time and huge memory? Does the computation time of DUBStepR include both regression, Guilt-by-association, and UMAP?

4) The confidence interval across seven datasets can be shown in Figure 3a.

5) In Figure 3b, it is recommended to use different colors for different datasets to show the datasets DUBStepR failed to be the rank 1, and which algorithm might supplement that situation

6) Figure 4a-b Density index label could be figure title to avoid overlap with a scatterplot.

7) Figure 5 as an overview of the workflow might be organized in the first section of the main text and Figure 1.

Reviewer #1 (Expertise: scRNASeq analysis, computational biology, ML):

In this paper, the authors present a new method for feature/gene selection in scRNA-seq data. This method is mainly based on gene-gene correlations, and is able to select features/genes that lead to improved performance in subsequent clustering analysis.

My major concerns about this paper are as follows:

1. According to the definition of correlation range score in the Method section, the max possible value for this score is 1.75. However, the boxplot in figure 1d showed scores exceeding 2. This does not seem to be correct.

We apologize for the lack of clarity on this point. The reviewer is correct in pointing out that, based on the formula in the Methods section, the maximum correlation range is indeed 1.75 ($1 - (-0.75)$). However, the metric being plotted in **Fig. 2d** (shown below) is not the correlation range. Rather, it's the correlation range score, which is the z-score of the correlation range (see Methods). Thus, the correlation range score can have values exceeding 2. To remove any ambiguity, we have changed the axis labels in **Fig. 2d-f** (shown below) to "Correlation Range Z-Score".

Fig. 2. Expression correlations of DE genes: scRNA-seq data from 5 lung adenocarcinoma cell lines (Tian et al., 2019). d) Boxplot showing correlation range scores for non-DE and DE genes. e-f) Scatter plot of genes showing correlation between e) $\log_2(\text{fold change})$ of cell-type-specific expression and f) $-\log_{10}(\text{q-value})$ of cell-type-specific expression with correlation range score. ****: p-value ≤ 0.0001 . rho: Spearman correlation.

2. The definition of range score seems to be problematic. Given this definition of $c_i = \max_3(G_i) - 0.75\min(G_i)$. Think about two hypothetical cases:

(1) If gene i 's correlation with all other genes are all 0.5, the range score will be $0.5 - 0.75 \cdot 0.5 = 0.125$.

(2) If gene i 's correlation with all other genes are all -0.5, the range score will be $-0.5 - 0.75 \cdot (-0.5) = -0.125$

First of all, it would be ideal that the range score for these two hypothetical cases are both 0. Second, even if not, the range score for these two hypothetical cases should at least be the same. More such make-up extreme examples can be made to show that such a definition can be problematic.

The reviewer is indeed correct that our correlation range treats positive and negative gene-gene correlations differently. In other words, we use a smaller coefficient for the most negative, i.e. lowest, correlation (0.75) than for the most positive, i.e. highest correlation (1.0). The rationale for this choice is that DE genes between cell types tend to show stronger positive than negative correlations. Below, we provide a scatterplot of maximum and minimum gene-gene correlations for DE genes in our benchmark scRNA-seq datasets (**Fig. R1.1**). While the distribution of max and min correlations does vary across datasets, we observe that a model with a smaller coefficient on min correlations ($\min(G_i) = -0.75 \cdot \max_3(G_i)$) provides a better fit to the DE gene data, on average, than a model with equal coefficients ($\min(G_i) = -\max(G_i)$). For this reason, we defined the correlation range in a manner that weights positive correlations more strongly than negative correlations.

Fig. R1.1. Scatterplots of minimum correlation vs maximum correlation of DE genes in the benchmarking datasets used in this manuscript. Blue line indicates the linear fit. m : slope.

The above scatter plots also illustrate the point that, in practice, the max and min correlations are not the same for any gene. Consider a hypothetical gene i that is positively correlated with all other genes. In such a scenario, when gene i has high expression in a cell, all other genes will also be likely to have high expression in the same cell. Conversely, in cells where gene i is low, all other genes will also tend to be low. This means that some cells will have uniformly high expression of all genes, while other cells have uniformly low expression of all genes. Such a scenario cannot exist, because single cell gene expression matrices are typically normalized so that the summed expression across genes is the same for all cells.

3. Back to Figure 1D, it shows that the smallest correlation range value can be lower than -1. I really cannot imagine an example that will lead to a -1 score, according to the range score definition.

Once again, we apologize for the lack of clarity on this point. As mentioned in our response to the reviewer comment 1, we are plotting the z-score of correlation range values, which can have values below -1.

4. In the stepwise regression procedure, the choice of a total of 100 steps is not well justified. In the method section, there is an assumption that the first 30 steps and the next 70 steps explain the same amount of variance. This is likely dataset-dependent. Since the total number step affects the identification of the elbow, the choices made here need to be properly evaluated and justified.

As noted by the reviewer, DUBStepR assumes that each individual step from Step 31 to Step 100 explains the same amount of variance as Step 30. This allows us to speed up the calculation: we only need to explicitly calculate the variance explained by the first 30 steps. We agree with the reviewer that the choice of number of steps needs to be properly evaluated, and could affect the location of the elbow. We have now systematically tested a range of steps for which variance was explicitly calculated from 20 to 100. Our results indicate that the location of the elbow remains unchanged from 30 to 100 steps in 6/7 benchmark datasets (**Supp. Table. S1**; see below). These results suggest that our choice of 30 steps provides an acceptable trade-off between computation time and accuracy of elbow point detection. We have now added this text to the Methods section:

“As illustrated in **Fig. 3i** of the revised manuscript (shown below), we explicitly calculate the variance explained by the first 30 steps, and then assume that the next 70 steps explain the same amount of variance as the 30th step, giving a total of 100 steps. We then draw the line (red in **Fig. 3i** of the revised manuscript) between the first and 100th step, and identify the elbow as the point on the variance explained

curve with the largest distance from this line. We observed that, while this procedure reduced compute time, it had little or no impact on the location of the elbow point (**Supp. Table S1**).”

Fig. 3. i) Scree plot: variance in GGC matrix explained by the gene regressed out at each step.

Dataset	20 steps	30 steps	50 steps	100 steps
FACS_PBMC	13	13	13	13
CRC_Cell_Line	19	22	22	22
3cl_10x	18	22	22	15
3cl_dropseq	14	14	14	14
3cl_celseq	12	11	11	11
5cl_10x	18	21	21	21
5cl_celseq	18	18	18	18

Supp. Table S1. Location of elbow point when regression is explicitly calculated for 20, 30, 50 and 100 steps.

5. With respect to cell clustering, the 7 scRNA-seq datasets benchmarked here are all relative easy datasets. 6 of the 7 are cells from multiple cell lines, which are easy to cluster. The last one of the 7 contains cells from FACS sorted subset of PBMCs, which is also easy to cluster because confusing/challenging cells in-between well established cell types/states are not going to be in the FACS gates. Therefore, the comparisons of algorithms are all done on easy datasets. Analysis based on datasets with "normal" level of difficulty will make the comparison more convincing.

We agree with the reviewer - this is indeed a major challenge for the single cell field. For quantitative benchmarking, we would ideally have access to the "true" (independently ascertained using a different modality) cell type labels for scRNA-seq datasets from complex primary samples. These true labels could then be used to quantify cell type separation using feature sets defined by various feature selection algorithms. Unfortunately, independently ascertained ground-truth cell type labels are not available in practice for the overwhelming majority of published scRNA-seq datasets.

Given the above, the field has adopted two distinct strategies for benchmarking scRNAseq algorithms. The first strategy involves simply downloading the cell type labels assigned by the data generators and assuming that they are accurate. The drawback of this approach is that these labels are typically generated by the authors using the very algorithms we are attempting to benchmark. In other words, there is an element of circularity in using the output of machine learning algorithms as ground truth to benchmark the same algorithms. It is precisely for this reason, i.e. the lack of independent ground truth standards, that Tian et al. and Zheng et al. generated the reference scRNA-seq datasets we have used for benchmarking in our study. We genuinely appreciate the limitation noted by the reviewer, namely that these reference datasets may not be as challenging to cluster as data from complex primary tissues. Nevertheless, we note that they have been widely adopted by the community for benchmarking purposes (Hou et al., 2020; Abdelaal et al., 2019; Wang et al., 2017; Li et al., 2018; Liu et al., 2020), since they have the key advantage of avoiding circularity.

To address the reviewer's comment regarding performance of feature selection methods on datasets that are more "normal," we generated scRNA-seq data from 8,312 unsorted PBMCs collected from four rheumatoid arthritis (RA) patients. To complement our previous quantitative benchmarking analyses, we used this dataset to qualitatively compare the results from the 8 feature selection methods. Remarkably, DUBStepR detected four unexpected rare subtypes and a coherent group of disease-associated T and NK cell states that were either undetected or only partially detected by the other feature selection methods. The corresponding manuscript text and figures are excerpted below.

“The above quantitative benchmarking analyses were largely based on detection of common cell types (>10% of all cells) in cell lines or FACS-purified cell populations from healthy donors. To demonstrate the ability of DUBStepR to cluster cells from a complex primary sample, we generated scRNA-seq data from 8,312 PBMCs from 4 rheumatoid arthritis (RA) patients. In this case, since the "true" cell type labels were unknown, our objective was to qualitatively compare results from the various feature selection methods. We used SingleR (Aran et al., 2019) to select the T and NK cell subset (5,329 cells) since this cell population is challenging to sub-cluster by conventional methods, despite its relevance to inflammatory phenotypes. DUBStepR (with DI optimization) identified 10 discrete subtypes in this dataset, with sharply distinct gene expression signatures (**Fig. 6a; Supp. Fig. S7**). These included four rare cell clusters that were undetected or only partially detected by the other feature selection methods: red blood cells (RBCs, 1.8%), proliferating cells (2%), platelet-T doublets (3.4%) and platelet-NK doublets (3%) (**Fig. 6b; Supp. Fig. S7, S8**). While RBCs reflect contamination during PBMC isolation, platelet-lymphocyte complexes and proliferating T cells regulated by *KIAA0101* are thought to play a role in the pathophysiology of RA (Zamora et al., 2013; Zamora et al., 2017; Aterido et al., 2014) (**Supp. Fig. S7**).

In addition to detecting multiple rare cell types, DUBStepR identified a dichotomy in CD4+ T, CD8+ T and NK cells, defined by coordinated differential expression of *SET*, *C1orf56*, *C16orf54*, *CDC42SE1* and *HNRNPH1* (**Supp. Fig. S7**), all of which have been previously identified as markers of a latently infected T cell subtype in HIV (Bradley et al., 2018). Once again, DUBStepR was the only feature selection method to clearly distinguish these cell states (**Supp. Fig. S7, S8**). In summary, DUBStepR was the only feature selection algorithm that robustly detected common and rare cell types and subtypes in this complex primary lymphocyte population.”

Fig. 6. Analysis of lymphocyte population from rheumatoid arthritis patient PBMCs. Left Panel: UMAP visualization of clustered identified by DUBStepR. Rest of the panels: UMAP visualization of cell clusters identified by DUBStepR using features selected by the 7 other feature selection methods.

Supp. Fig. S7. Feature selection on rheumatoid arthritis T and NK cells using DUBStepR. a) Feature plots demonstrating expression of markers. b) Expression of *C1orf56* and its correlated cell state markers. Related to Fig. 6.

Supp. Fig. S8. Feature selection on rheumatoid arthritis T and NK cells using HVGST. a) Feature plots demonstrating expression of markers. b) Expression of *C1orf56* and its correlated cell state markers. Related to Fig. 6.

6. The first paragraph in page 5, the statement of "For optimal cell type clustering, a feature selection algorithm should ideally select only cell-type-specific genes (DE genes) as features", and the evaluation figure 3c based on this statement, both are flawed. Yes, cell-type-specific genes (for the known cell types / cell lines) should be selected so that these cell types/lines can be separated. However, selecting other genes does not mean that the selection algorithm is bad. If one cell line has a lot of heterogeneity (several subtypes), a good feature selection algorithm should select

genes that can help cluster the subtypes within that cell line, but these genes are not "cell-type-specific genes" in this evaluation. Therefore, the proposed method winning this evaluation shown in Figure 3c does not really provide a convincing argument that the proposed method is better. It is actually misleading.

We thank the reviewer for pointing out this issue. We have revised the text in the manuscript to refer to "genes specific to cell types or subtypes".

Regarding the validity of the benchmarking in **Fig. 4** (Fig. 3 in the original manuscript), we note that these datasets have been widely used in the field for benchmarking purposes (for example: Hou et al., 2020; Abdelaal et al., 2019; Wang et al., 2017; Li et al., 2018; Liu et al., *Nat Commun*, 2020) because the true cell type labels are known. We used the cell type labels assigned by the original authors as ground truth, and all other benchmarking studies have done the same.

We agree with the reviewer that there could be further heterogeneity (subtypes) within each cell type. However, the ground-truth labels are not known for any such subtypes, and thus we have no way of quantifying the accuracy of subtype identification in these datasets. It is for this reason that benchmarking studies based on these datasets use the cell type labels assigned by the original authors.

7. The density index idea can also be problematic. If there exist a set of genes that leads to many many cell clusters and each cluster has more than k cells, the DI score will be good. However, such a set of genes certainly are not good features for clustering, and the SI score based on the known cell clusters is going to be poor. This is one counter example showing that the DI score may not be a good approximation of the SI score in certain scenarios. It is good that Figure 4c shows the correlation between SI and DI in the cell line data. But again, the datasets analyzed here are all easy datasets for cell clustering, limiting the generality of the conclusions.

As noted by the reviewer, the DI is designed to quantify the degree to which cells form dense clumps (aka clusters) in gene expression space. This follows the approach of the well-established DBSCAN clustering algorithm (Ester et al. (1996)), which defines clusters as regions in feature space with a high density of data points. The DI, as we have defined it, uses an efficient graph-based strategy to quantify the local density of data points, and is thus similar in spirit to the explicit density calculation used by DBSCAN.

The reviewer notes that it is possible for a feature set to divide cells into a large number of well-separated clusters, many or most of which might contain only a small number of cells. This is indeed true. The reviewer then goes on to suggest that such a clustering would be self-evidently incorrect. Here we respectfully disagree. As we show with new data in this revision, DUBStepR selected a feature set for PBMCs

from RA patients that created multiple fragmentary clusters not well detected by existing feature selection methods. However, these clusters were not artifacts of the DI. Rather, they represented genuine cell types and subtypes that were consistent with the literature and had independent support from previous studies (**Fig. 6**).

In addition to the above-mentioned RA dataset, we have used DUBStepR on diverse primary sample scRNA-seq datasets in the course of ongoing studies (unpublished, in prep): PBMCs from COVID-19 patients, PBMCs from patients with type 1 diabetes, colorectal tumors, bone marrow from healthy donors and those with chronic myeloid leukemia, human fetal brain, mouse and rat brain and human brain organoids. We did not detect artifactual fragmentation in the DUBStepR results in any of these datasets.

8. The analysis/comparison on scATACseq data in Figure 7 is not sufficiently described. Looking at Figure 7, it is unclear what the figure shows the proposed feature selection is better than not doing feature selection at all. More discussions/interpretations are needed to make this section meaningful.

We thank the reviewer for raising this concern. We have updated **Fig. 7** and its corresponding text to include Highly Variable Features (HVF) - an extension of the popular HVG approach implemented in Seurat. In the updated **Fig. 7**, we plotted the graph topology of the Monocle3 trajectories generated using DUBStepR, HVF and without feature selection. We then compared them to the hematopoietic differentiation schematic provided by Buenrostro et al. (Cell, 2018), from where the dataset was also obtained. The updated manuscript text and figure are presented below.

“Feature selection is typically not performed on scATAC-seq data, since their almost binary nature (most genomic bins have zero or one count) renders them refractory to conventional single cell feature selection techniques based on variance-mean comparison (Stuart et al., 2020). However, since the logic of feature correlations applies even to binary or almost binary data, we hypothesized that DUBStepR could also improve the quality of cell type inferences from this data type. To test this hypothesis, we applied DUBStepR to scATAC-seq data from 8 FACS-purified subpopulations of human bone marrow cells (Buenrostro et al., 2018). In contrast to the common approach of using all scATAC-seq peaks, we found that peaks selected by DUBStepR more clearly revealed the emergence of the three major lineages from hematopoietic stem cells: lymphoid, myeloid and megakaryocyte/erythroid (**Fig. 7a-c**). Specifically, trajectory analysis using Monocle 3 (Cao et al., 2019) yielded a topology that matched the known hematopoietic differentiation hierarchy (Buenrostro et al., 2018) (**Fig. 7g**) only in the case of DUBStepR (**Fig. 7d-f**).”

Fig. 7. Comparison of feature selection on scATAC-seq data. a-c) UMAP visualizations of cells colored by FACS-sorted cell type labels, with trajectories constructed using Monocle 3. a) All peaks (no feature selection), b) Peaks selected by Highly Variable Features (HVF), c) Peaks selected by DUBStepR, d-f) Monocle-generated trajectories on the corresponding UMAP visualizations. g) Hematopoietic differentiation schematic as described in Buenrostro et al. CLP: Common Lymphoid Progenitor, CMP: Common Myeloid Progenitor, GMP: Granulocyte/Macrophage Progenitor, HSC: Haematopoietic Stem Cell, LMPP: Lymphoid-primed Multipotent Progenitor, MEP: Megakaryocyte/Erythroid Progenitor, Mono: Monocyte, MPP: Multipotent Progenitor

Overall, multiple aspects of this paper seem to be problematic to me. Therefore, I am afraid that I cannot give a positive recommendation for this paper.

We thank the reviewer for helping us strengthen the manuscript, and hope the revised version will be considered more favourably.

Reviewer #2 (Expertise: Epigenetics, single cell methods, computational biology):

The authors describe a novel and effective approach for the feature selection method in scRNA-seq data to improve single cell clustering. This is an important problem in the field that could be widely applied for aiding insights from sparse complex multidimensional data.

DUBStepR first carries out feature selection based on the idea that top differential genes of same cell types correlate highly with each other, to generate a gene-gene correlation matrix. An optimal number of feature genes is then determined, which is very important since if it is too high, the separation of cells would not be optimal and if it is too few, it may not work well due to the sparse nature of the data, both leading to sub-optimal clustering. Often researchers need to try different number of genes to use (highly variable genes being one of the most common method currently used) and decide based on the outcome, which tends to be subjective. Thus, the approach DUBStepR takes to determine the optimal number of feature genes is highly appreciated.

The authors show that DUBStepR outperforms compared to other existing methods by multiple measures. Excitingly, although DUBStepR was not originally developed to detect rare cell types, they have shown it is also promising in this regard. Furthermore, DUBStepR appears to perform well with scATAC-seq datasets and can in principle be applied to other single-cell epigenomic datasets such as scChIP-seq and DNA methylome datasets. The authors should be praised for providing all code and data in publicly available repositories and implementing the tool as a (creatively named) R package to facilitate wider use.

Overall it is a very well-written paper with some interesting mathematical insights. DUBStepR is worth bringing attention of to the community of single cell genomics and recommend it to be published upon taking in the points below.

We are tremendously grateful for the reviewer's comments and do believe that DUBStepR will be a useful tool for the single cell community.

Major comments:

I would like to see how much better the DUBStepR performs compared to other methods on a more complex sample that contains many more cell types than those that have been used in the manuscript, such as mouse lung. Mouse lung would also be a good tissue to test whether the DUBStepR would detect rare cell types, as claimed by the authors, such as ILC2s. ILC2s are present in the lung but not

detected with currently available data processing tools due to its low number in steady state. The single cell genomics community would appreciate to see this.

We greatly appreciate this comment - a similar point has also been raised by the other 2 reviewers. Rare cell type detection is indeed a major challenge for the single cell field. For quantitative benchmarking, we would ideally have access to the “true” (independently ascertained using a different modality) cell type labels for scRNA-seq datasets from complex primary samples. These true labels could then be used to quantify cell type separation using feature sets defined by various feature selection algorithms. Unfortunately, independently ascertained ground-truth cell type labels are not available in practice for the overwhelming majority of published scRNA-seq datasets. Mouse lung is a case in point - this is a complex tissue with a large number of cell types and subtypes that are still being mapped. We are unfortunately not aware of a mouse lung scRNA-seq dataset for which the true cell type labels have been independently determined, which we could use as a benchmark for quantifying cell type separation. In addition, since the ILC2 cell type has been discovered in the single-cell RNA sequencing modality using Seurat’s Highly Variable Genes (HVG) feature selection approach (Wallrapp et al., 2017; Ghaedi et al., 2020), this cell type may not be ideal for an unbiased comparison of HVG with other feature selection methods.

To address the comment above as well as comments from the other 2 reviewers regarding performance of feature selection methods on data from complex primary samples, we generated scRNA-seq data from 8,312 unsorted PBMCs collected from four rheumatoid arthritis patients. To complement our previous quantitative benchmarking analyses, we used this dataset to qualitatively compare the results from the 8 feature selection methods. Remarkably, DUBStepR detected four unexpected rare subtypes and a coherent group of disease-associated T and NK cell states that were either undetected or only partially detected by the other feature selection methods. The corresponding manuscript text and figures are excerpted below.

“ To demonstrate the ability of DUBStepR to cluster cells from a complex primary sample, we generated scRNA-seq data from 8,312 PBMCs from 4 rheumatoid arthritis (RA) patients. In this case, since the “true” cell type labels were unknown, our objective was to qualitatively compare results from the various feature selection methods. We used SingleR (Aran et al., 2019) to select the T and NK cell subset (5,329 cells) since this cell population is challenging to sub-cluster by conventional methods, despite its relevance to inflammatory phenotypes. DUBStepR (with DI optimization) identified 10 discrete subtypes in this dataset, with sharply distinct gene expression signatures (**Fig. 6a; Supp. Fig. S7**). These included four rare cell clusters that were undetected or only partially detected by the other feature selection methods: red blood cells (RBCs, 1.8%), proliferating cells (2%), platelet-T doublets (3.4%) and platelet-NK doublets (3%) (**Fig. 6b; Supp. Fig. S7, S8**). While RBCs

reflect contamination during PBMC isolation, platelet-lymphocyte complexes and proliferating T cells regulated by *KIAA0101* are thought to play a role in the pathophysiology of RA (Zamora et al., 2013; Zamora et al., 2017; Aterido et al., 2014) (**Supp. Fig. S7**).

In addition to detecting multiple rare cell types, DUBStepR identified a dichotomy in CD4+ T, CD8+ T and NK cells, defined by coordinated differential expression of *SET*, *C1orf56*, *C16orf54*, *CDC42SE1* and *HNRNPH1* (**Supp. Fig. S7**), all of which have been previously identified as markers of a latently infected T cell subtype in HIV (Bradley et al., 2018). Once again, DUBStepR was the only feature selection method to clearly distinguish these cell states (**Supp. Fig. S7, S8**). In summary, DUBStepR was the only feature selection algorithm that robustly detected common and rare cell types and subtypes in this complex primary lymphocyte population.”

Fig. 6. Analysis of lymphocyte population from rheumatoid arthritis patient PBMCs. Left Panel: UMAP visualization of clustered identified by DUBStepR. Rest of the panels: UMAP visualization of cell clusters identified by DUBStepR using features selected by the 7 other feature selection methods.

Supp. Fig. S7. Feature selection on rheumatoid arthritis T and NK cells using DUBStepR. a) Feature plots demonstrating expression of markers. b) Expression of *C1orf56* and its correlated cell state markers. Related to **Fig. 6**.

Supp. Fig. S8. Feature selection on rheumatoid arthritis T and NK cells using HVGST. a) Feature plots demonstrating expression of markers. b) Expression of *C1orf56* and its correlated cell state markers. Related to **Fig. 6**.

Some of the single-cell techniques demonstrated (10x Genomics, Drop-seq) use UMI counts whereas others (Smart-Seq2) use read counts. Does feature selection work better with UMI or can it be applied to either dataset?

Feature selection has been applied to read count data ever since the development of the Highly Variable Genes method in 2013 (Brennecke et al., 2013; Kim and Marioni, 2013; Kolodziejczyk et al., 2015; Buettner et al., 2015). More recently, with the

prevalence of UMI-based data, HVG has also become the default method for this data type. It is challenging to quantitatively compare feature selection results on read count vs UMI data, because we are not aware of a benchmark dataset with independently ascertained cell type labels that includes both read count and UMI data generated side-by-side from the same samples.

In this study, we have benchmarked DUBStepR and other feature selection methods on read count data (*CRC_Cell_Line* dataset; Li et al., 2017) as well as UMI data (all other datasets).

Several claims are made regarding scATAC-seq data. How does step-wise regression on a gene-gene correlation matrix work differently on a binary scATAC or scChIP peak count matrix? Would you expect it to work similarly on Beta-values for single-cell bisulfite sequencing for DNA methylation? More supporting data to back up the claims about epigenomics applications would be helpful.

We appreciate these questions raised by the reviewer. The scATAC-seq analysis we describe in the manuscript uses peaks as features, and identifies correlated peaks on the peak count matrix. Thus, we expect DUBStepR to perform similarly on an scChIP-seq peak count matrix, and with beta values from single-cell bisulphite sequencing. We have not yet tested DUBStepR on binary matrices, since binary matrices could be more amenable to other gene-gene similarity metrics such as the Jaccard index (for example). We hope to extend DUBStepR to such datasets soon.

We are very interested in expanding DUBStepR to tackle additional single-cell epigenomics modalities. However, the absence of benchmarking datasets for scChIP and single-cell bisulfite sequencing makes it challenging to assess performance improvements provided by DUBStepR, and thus we look forward to addressing this concern as the single cell epigenomics field expands and ground truth cell type labels become available.

For the scATAC-seq dataset, I am curious what peaks are selected by DUBStepR. Are they mostly peaks associated with coding or non-coding genes? Promoters or distal? Since enhancers are known to be more cell type-specific, overview of such categories of the peaks that were selected by DUBStepR would be of interest to the reviewers.

Using the annotated BED file provided by Buenrostro et al. and the Ensembl-based annotation database (Ensembl.Hsapiens.v86), we found only a minor increase in association with protein coding genes of peaks selected by DUBStepR as compared to all peaks (76.06% versus 74.52%; **Fig. R2.1a**). To determine whether the peaks corresponded to proximal or distal elements, we used the *chipenrich* package (Welch et al., 2014) to plot the distribution of peak distances to the nearest TSS. Within a 5 kb window from the TSS, we observed a modest cumulative increase in

enrichment of about 4.1% in the DUBStepR-selected peaks compared to all peaks (Fig. R2.1b). In summary, DUBStepR-selected peaks did not demonstrate specific enrichment of an associated gene type, nor did they contain a substantial excess of proximal or distal elements.

Fig. R2.1. a) Distribution of the type of gene associated with peaks selected by DUBStepR versus all peaks. b) Distribution of distance to the nearest TSS from peaks selected by DUBStepR as compared to all peaks.

Several claims of computational performance are made but DUBStepR appears to have similar or greater cpu-time and memory requirements to other techniques (Fig. S3). Is this acceptable given the higher accuracy from iterative step-wise regression?

The reviewer raises an important question in this comment. DUBStepR first computes the gene-gene correlation matrix and then performs all subsequent operations on this matrix, whose size is independent of the number of cells. The algorithm is thus more resilient to the exponentially increasing number of cells in single-cell experiments. We observe this scalability of DUBStepR in our benchmarking analysis:

“DUBStepR used 6.4x more memory to process the 1M cell dataset as compared to the 100k dataset. In contrast, HVGDisp, HVGVST, trendVar, devianceFS and M3DropDANB all increased their memory consumption by ~12.5x.”

We also note that, since HVG was first introduced in 2013 (Brennecke et al., 2013), its software implementation is mature and close to optimal. We expect further scalability improvements to DUBStepR in subsequent versions (for instance, by implementing base functions in C/C++).

The results from the computational performance benchmarking in the manuscript and related figure are presented below:

“DUBStepR, HVGDisp, HVGST, trendVar, devianceFS and M3DropDANB were able to process the entire 1 million cell dataset, while GiniClust and HLG could not scale to 100k cells (**Supp. Fig. S6**). On the largest dataset, DUBStepR ranked fourth out of the eight tested methods in memory consumption and compute time. In terms of memory scalability, DUBStepR used 6.4x more memory to process the 1M cell dataset as compared to the 100k dataset. In contrast, HVGDisp, HVGST, trendVar, devianceFS and M3DropDANB all increased their memory consumption by ~12.5x. Thus, DUBStepR is scalable to over a million cells and shows promise for even larger datasets.”

Supp. Fig. S6. Benchmarking of computational efficiency of DUBStepR against existing feature selection methods on datasets of 10k, 100k and 1million cells. a) Execution time (in minutes) taken by each method. b) Total memory consumed (in GB) by each method. The X-axes of both plots were log-transformed for ease of visualization.

Minor points:

How does DUBStepR remove mitochondrial and ribosomal genes? Is this based on gene ID or symbols? If it is hardcoded for the human data demonstrated, is it possible to apply to other species with different gene naming conventions?

We thank the reviewer for this important question, which would be very pertinent to users of DUBStepR. DUBStepR identifies and removes mitochondrial and ribosomal genes using gene symbols and Ensembl IDs in human, mouse and rat data. We have now modified the Methods section to clarify this, and copied the text below:

“Additionally, mitochondrial genes, ribosomal genes, and pseudogenes are identified using gene symbols or Ensembl IDs for human, mouse and rat datasets using the latest Ensembl references downloaded from BioMart (Howe et al., 2021).”

If 30 steps are performed as an approximation of 100 steps, it seems unnecessary to mention that 100 steps are used. The number of iterations appears to have been reduced for performance. This could be stated more unequivocally.

We apologize for the lack of clarity in explaining the elbow point detection procedure. We have now expanded the text in the methods section to address this point (**Supp. Table S1**, see response to Reviewer 1 comment 4 above). The corresponding manuscript text, figure and table are copied below:

“As illustrated in **Fig. 3i** of the revised manuscript (shown below), we explicitly calculate the variance explained by the first 30 steps, and then assume that the next 70 steps explain the same amount of variance as the 30th step, giving a total of 100 steps. We then draw the line (red in **Fig. 3i** of the revised manuscript) between the first and 100th step, and identify the elbow as the point on the variance explained curve with the largest distance from this line. We observed that, while this procedure reduced compute time, it had little or no impact on the location of the elbow point (**Supp. Table S1**).”

Fig. 3. i) Scree plot: variance in GGC matrix explained by the gene regressed out at each step.

Dataset	20 steps	30 steps	50 steps	100 steps
FACS_PBMC	13	13	13	13
CRC_Cell_Line	19	22	22	22
3cl_10x	18	22	22	15
3cl_dropseq	14	14	14	14
3cl_celseq	12	11	11	11
5cl_10x	18	21	21	21
5cl_celseq	18	18	18	18

Supp. Table S1. Location of elbow point when regression is explicitly calculated for 20, 30, 50 and 100 steps.

It would be helpful to distinguish datasets in Fig 3b (by colors, shapes, or numeric text labels).

We thank the reviewer for this suggestion. We have updated Fig. 3 (now **Fig. 4** in the revised manuscript) as recommended, with different colors representing the different datasets, as depicted below:

Fig. 4. Benchmarking feature selection methods. a) Mean scaled Silhouette Index of feature sets ranging from 50 to 4,000 features. b) Rank distribution of feature selection methods. For each dataset, the 7 methods are ranked from 1 to 7 by their best SI across all feature set sizes. c) AUROC of DE gene detection. *: $p \leq 0.05$, **: $p \leq 0.01$, ***: $p \leq 0.001$.

The R package is now available on CRAN and the manuscript should be updated to reflect this.

We appreciate this point from the reviewer. We have updated the manuscript to reflect the DUBStepR CRAN package.

Reviewer #3 (Expertise: scRNASeq analysis, scATAC-Seq, ML, computational biology):

Clustering analysis is a key component in the computational pipeline of Single-cell RNA-seq and ATAC-seq to understand the cellular heterogeneity and extract cell type/state markers. One of the important prior steps before clustering is to determine a suitable top variable feature (gene) set that largely captures differential expressed genes across cell types. In this study, the authors present a new framework based on gene expression correlation to select an optimal set of features by an optimized stepwise correlation, guilt-by-association, and newly invented density index (DI). They benchmarked the framework across the different platforms to validate its significant improvement on SI and DI scores than other software. The R package is fully integrated into the classic Seurat workflow and might be promising to become a routine replacement of HVGST in the future. However, the manuscript still needs to be largely improved to make the method and the validation more clear and solid, especially for the real biological datasets in the case studies. Please see the following comments.

We greatly appreciate the comments and suggestions below, which have helped us strengthen the manuscript. We hope the methodology and validation have been clarified to the satisfaction of the reviewer, particularly with the inclusion of a new scRNA-seq dataset from rheumatoid arthritis PBMC samples, and that the revised version will be received more favourably.

Major comments:

1) The max3 function in the method part is not clear. Please provide the mathematical formula to define it.

We apologize to the reviewer for this lack of clarity. We have updated the text in the Methods section to address this comment, and added the mathematical formula to define it. The updated text is presented below.

“Correlation range c_i for gene i can be defined for a gene-gene correlation matrix G as

$$c_i = \max_3(G_i) - 0.75 * \min(G_i),$$

where $\max_h(G_i)$ refers to the h^{th} -largest correlation value in column i of G . Correlation range uses the second-largest non-self correlation value (3rd largest value in GGC

column) to calculate the range, so as to protect against genes with overlapping 5' or 3' exons (Nakayama et al., 2007).”

2) In a routine clustering or trajectory analysis, we usually select a subset of 1000~2000 genes. In Figure 3a, when selecting 2000 features, the SI increase of DUBStepR is minor in comparison to Seurat and HLG methods, but the DE metrics in Figure 3c is quite significant, given that in Figure 1, the correlation between correlation range score and log fold change is highly significant, is there any mathematical link between the computation of correlation range score and the method of differential gene expression analysis?

We thank the reviewer for raising this interesting question. While DE genes tend to have significantly higher correlation range z-scores than non-DE genes (Fig. 2d of the revised manuscript) and we observe an empirical correlation between the correlation range z-score and differential expression fold change and q-value (Fig. 2e-f of the revised manuscript), the computation of the correlation range z-score is completely agnostic to cell type labels and has no obvious mathematical link to differential expression analysis. Further studies will be needed to investigate this relationship in a more formal manner. The referenced figures are copied below.

Fig. 2. Expression correlations of DE genes: scRNA-seq data from 5 lung adenocarcinoma cell lines (Tian et al., 2019). d) Boxplot showing correlation range scores for non-DE and DE genes. e-f) Scatter plot of genes showing correlation between e) $\log_2(\text{fold change})$ of cell-type-specific expression and f) $-\log_{10}(\text{q-value})$ of cell-type-specific expression with correlation range score. ****: p-value ≤ 0.0001 . rho: Spearman correlation.

3) Please consider adding a state-of-art method deviance score and Pearson residual from GLM-PCA study and associated scry R package for feature selection of UMI-based single-cell RNA-seq, and provide corresponding SI and DI metrics in Figure 3a-c.

We have now included the deviance score-based feature selection from the GLM-PCA study, through the *scry* R package in our benchmarking analysis. DUBStepR remains the top-performing method in terms of SI as well as AUROC of DE gene detection, and our conclusions remain unchanged.

The updated benchmarking results are presented below, along with **Fig. 4** of the revised manuscript:

“In contrast to DUBStepR, which showed maximal performance at 200-300 features, the other methods remained close to their respective performance peaks over a broad range from 300 to 2,000 features and dropped off on either side of this range. DUBStepR substantially outperformed all other methods across the entire range of feature set size (**Fig. 4a**). Moreover, DUBStepR was the top-ranked algorithm on 5 of the 7 datasets (**Fig. 4b**).

Remarkably, DUBStepR achieved an AUROC in excess of 0.97 on all 7 datasets, indicating near-perfect separation of DE and non-DE genes (**Fig. 4c**). devianceFS was able to exceed the same performance threshold on 4 of the 7 datasets and HLG on only one. All other methods demonstrated significantly lower performance (**Fig. 4c**). Thus, DUBStepR greatly improves our ability to select cell type/subtype specific marker genes (DE genes) for clustering scRNA-seq data.”

Fig. 4. Benchmarking feature selection methods. a) Mean scaled Silhouette Index of feature sets ranging from 50 to 4,000 features. b) Rank distribution of feature selection methods. For each dataset, the 7 methods are ranked from 1 to 7 by their best SI across all feature set sizes. c) AUROC of DE gene detection. *: $p \leq 0.05$, **: $p \leq 0.01$, ***: $p \leq 0.001$.

4) In the section of “Guilt-by-association expands the feature set”, authors claim each individual signature is represented by 2-5 genes, but Fig.2i select the elbow point by 20 genes. What does the signature mean here? Is it a cell cluster signature?

We apologize for the lack of clarity in this point. We have now added text to indicate that, in this context, the word “signature” is used to mean a set of correlated genes. In other words, it is indeed akin to a cell cluster signature. **Fig. 3j** (in the revised manuscript; shown below) demonstrates the set of 20 “seed” genes, which represent the various signatures (patterns) of expression in the dataset. For instance, genes *ALDH1A1* and *KRT81* demonstrate an expression signature specific to the A549 cell cluster (shown below).

Fig. 3. j) Standardized average expression of the final seed gene set in each of the 5 cell lines.

Furthermore, DUBStepR detects a handful of seed genes in backward stepwise regression and expands the seed genes by recycling correlated genes from the candidate feature set in a forward way, could the two steps combined together to select the feature set once by a certain regression statistic such as BIC?

We thank the reviewer for this suggestion. Indeed, BIC is commonly used for selecting the optimal number of regression steps. When we initially designed

DUBStepR, we expected, as suggested by the reviewer, that there would be an optimal number of regression steps, and that if we stopped the regression at the right point, we would directly obtain the optimal feature set. However, in practice, we found that, no matter how many steps we used in the regression, we could still improve the clustering results by adding genes to the feature set in a forward manner using guilt-by-association. DUBStepR therefore uses a two-stage procedure for feature selection.

5) The description of Guilt-by-association is too brief in the main text to understand and ignored in the Methods section. From the text and Figure 5, it is hard to tell how to expand the feature set on the genes UMAP, is it conducted by finding the nearest genes of seed genes? The authors need to describe more details about this step. DUBStepR orders genes first by Guilt-by-association order, then correlation range. What is the metric output by the Guilt-by-association step? Is the Guilt-by-association metric more informative than the stepwise regression and correlation range?

We are grateful to the reviewer for raising this important point, and apologise for omitting a complete explanation of the guilt-by-association step in the manuscript.

We have added a new subsection in the Methods section dedicated to guilt-by-association, and have expanded the results section as well. The manuscript text is copied below.

“Guilt-by-Association

Guilt-by-association, also known as label propagation through a network, allows DUBStepR to determine a robust ordering of features in an iterative manner. Once the seed genes have been determined, the gene with the strongest Pearson correlation to any of the seed genes is first identified. This gene is then added to the seed genes, thereby expanding the feature set. This feature set (now consisting of the seed genes and the newly added feature gene) is then used to identify the next most strongly correlated gene, which is again added to the feature set. By iteratively repeating the latter step, DUBStepR propagates through the gene-gene correlation network until the feature set has reached its final size.”

6) DUBStepR application in sorted PBMC dataset is not a good example, since Basophil cell population is constituted with only 12 cells and the SI is equal to Seurat HVG VST, it is not superior to use DUBStepR to identify five additional cells than the routine Seurat workflow, authors may need to find a better real biological dataset to justify the clustering improvement of DUBStepR to identify rare cell type.

We appreciate this comment from the reviewer. To strengthen the conclusion regarding rare cell type detection, we have now generated a real biological dataset as requested by the reviewer (as well as the other 2 reviewers). We used scRNA-seq to profile 8,312 unsorted PBMCs collected from four rheumatoid arthritis patients. To complement our previous quantitative benchmarking analyses, we used this dataset to qualitatively compare the results from the 8 feature selection methods. Remarkably, DUBStepR detected four unexpected rare subtypes and a coherent group of disease-associated T and NK cell states that were either undetected or only partially detected by the other feature selection methods. The corresponding manuscript text and figures are excerpted below.

“The above quantitative benchmarking analyses were largely based on detection of common cell types (>10% of all cells) in cell lines or FACS-purified cell populations from healthy donors. To demonstrate the ability of DUBStepR to cluster cells from a complex primary sample, we generated scRNA-seq data from 8,312 PBMCs from 4 rheumatoid arthritis (RA) patients. In this case, since the "true" cell type labels were unknown, our objective was to qualitatively compare results from the various feature selection methods. We used SingleR (Aran et al., 2019) to select the T and NK cell subset (5,329 cells) since this cell population is challenging to sub-cluster by conventional methods, despite its relevance to inflammatory phenotypes. DUBStepR (with DI optimization) identified 10 discrete subtypes in this dataset, with sharply distinct gene expression signatures (**Fig. 6a; Supp. Fig. S7**). These included four rare cell clusters that were undetected or only partially detected by the other feature selection methods: red blood cells (RBCs, 1.8%), proliferating cells (2%), platelet-T doublets (3.4%) and platelet-NK doublets (3%) (**Fig. 6b; Supp. Fig. S7, S8**). While RBCs reflect contamination during PBMC isolation, platelet-lymphocyte complexes and proliferating T cells regulated by *KIAA0101* are thought to play a role in the pathophysiology of RA (Zamora et al., 2013; Zamora et al., 2017; Aterido et al., 2014) (**Supp. Fig. S7**).

In addition to detecting multiple rare cell types, DUBStepR identified a dichotomy in CD4⁺ T, CD8⁺ T and NK cells, defined by coordinated differential expression of *SET*, *C1orf56*, *C16orf54*, *CDC42SE1* and *HNRNPH1* (**Supp. Fig. S7**), all of which have been previously identified as markers of a latently infected T cell subtype in HIV (Bradley et al., 2018). Once again, DUBStepR was the only feature selection method to clearly distinguish these cell states (**Supp. Fig. S7, S8**). In summary, DUBStepR was the only feature selection algorithm that robustly detected common and rare cell types and subtypes in this complex primary lymphocyte population.”

Fig. 6. Analysis of lymphocyte population from rheumatoid arthritis patient PBMCs. Left Panel: UMAP visualization of clustered identified by DUBStepR. Rest of the panels: UMAP visualization of cell clusters identified by DUBStepR using features selected by the 7 other feature selection methods.

Supp. Fig. S7. Feature selection on rheumatoid arthritis T and NK cells using DUBStepR. a) Feature plots demonstrating expression of markers. b) Expression of *C1orf56* and its correlated cell state markers. Related to Fig. 6.

Supp. Fig. S8. Feature selection on rheumatoid arthritis T and NK cells using HVGST. a) Feature plots demonstrating expression of markers. b) Expression of *C1orf56* and its correlated cell state markers. Related to Fig. 6.

7) *It is not clear that DUBStepR surpasses other methods in Figure 7. Are the dimensional reduction and clustering of single-cell ATAC-seq based on the k-mers, peak counts, or gene scores? What's the SI and DI comparison among feature selection of DUBStepR and Seurat and all features based on the FACS-sorting information? Correlation measurement might be changed in DUBStepR since scATAC-seq is a binary matrix. Authors may consider the Jaccard index used by Snapatac as similarity measurement.*

We apologize for the lack of clarity regarding the improvement provided by DUBStepR in analyzing the scATAC-seq dataset (Fig. 7). The dimensionality reduction of scATAC-seq data was based on peak counts, and thus the matrix was not binary (Supp. Note 2D). We therefore used gene-gene correlations as before, rather than the Jaccard index used by SnapATAC. As recommended, we have augmented this result to compare DUBStepR to Seurat's Highly Variable Features (HVF) approach for feature selection. Cell labels were defined by FACS-sorting (Buenrostro et al., 2018).

The SI and DI metrics quantify the separation of discrete cell clusters in feature space. However, in contrast to the discrete, well-separated clusters typically observed in differentiated tissues, human bone marrow is characterized by a continuum of cell states within a differentiating population of cells. We therefore evaluated feature selection methods by comparing the resulting continuous differentiation trajectories to the known differentiation hierarchy of hematopoiesis in bone marrow, as summarized in the paper from which the dataset was obtained

(Buenrostro et al., 2018). The updated manuscript text and figure are excerpted below:

“Feature selection is typically not performed on scATAC-seq data, since their almost binary nature (most genomic bins have zero or one count) renders them refractory to conventional single cell feature selection techniques based on variance-mean comparison (Stuart et al., 2020). However, since the logic of feature correlations applies even to binary or almost binary data, we hypothesized that DUBStepR could also improve the quality of cell type inferences from this data type. To test this hypothesis, we applied DUBStepR to scATAC-seq data from 8 FACS-purified subpopulations of human bone marrow cells (Buenrostro et al., 2018). In contrast to the common approach of using all scATAC-seq peaks, we found that peaks selected by DUBStepR more clearly revealed the emergence of the three major lineages from hematopoietic stem cells: lymphoid, myeloid and megakaryocyte/erythroid (**Fig. 7**). Specifically, trajectory analysis using Monocle 3 (Cao et al., 2019) yielded a topology that matched the known hematopoietic differentiation hierarchy (Buenrostro et al., 2018) (**Fig. 7g**) only in the case of DUBStepR (**Fig. 7d-f**).”

Fig. 7. Comparison of feature selection on scATAC-seq data. a-c) UMAP visualizations of cells colored by FACS-sorted cell type labels, with trajectories constructed using Monocle 3. a) All peaks (no feature selection), b) Peaks selected by Highly Variable Features (HVF), c) Peaks selected by DUBStepR, d-f) Monocle-generated trajectories on the corresponding UMAP visualizations. g) Hematopoietic differentiation schematic as described in Buenrostro et al. CLP: Common Lymphoid Progenitor, CMP: Common Myeloid Progenitor, GMP: Granulocyte/Macrophage Progenitor, HSC: Haematopoietic Stem Cell, LMPP: Lymphoid-primed Multipotent Progenitor, MEP: Megakaryocyte/Erythroid Progenitor, Mono: Monocyte, MPP: Multipotent Progenitor

8) Would the feature selection proposed by DUBStepR improve another important downstream analysis dimensionality reduction such as PCA, UMAP, and TSNE? Are the UMAPs and clusters shown in the manuscript using the same feature selected in the DUBStepR?

Our Silhouette Index benchmarking (Fig. 4a,b in the revised manuscript) is performed in the space of principal components, and hence demonstrates that DUBStepR provides an improvement in principal component analysis (PCA). As the UMAP and tSNE visualizations are computed using the output of PCA, the improvement in PC-space by DUBStepR translates to an improvement in the UMAP/tSNE visualization. This is succinctly depicted in Fig. 6 & 7 of the revised manuscript, where cell clusters or lineages are more accurately reproduced by DUBStepR than by other feature selection strategies. Thus, performing feature selection using DUBStepR improves downstream dimensionality reduction in single-cell data. The UMAPs and clusters shown in the manuscript are computed on the features selected by DUBStepR. The referenced figures are copied below.

Fig. 4. Benchmarking feature selection methods. a) Mean scaled Silhouette Index of feature sets ranging from 50 to 4,000 features. b) Rank distribution of feature selection methods. For each dataset, the 7 methods are ranked from 1 to 7 by their best SI across all feature set sizes.

Fig. 6. Analysis of lymphocyte population from rheumatoid arthritis patient PBMCs. Left Panel: UMAP visualization of clustered identified by DUBStepR. Rest of the panels: UMAP visualization of cell clusters identified by DUBStepR using features selected by the 7 other feature selection methods.

Fig. 7. Comparison of feature selection on scATAC-seq data. a-c) UMAP visualizations of cells colored by FACS-sorted cell type labels, with trajectories constructed using Monocle 3. a) All peaks (no feature selection), b) Peaks selected by Highly Variable Features (HVF), c) Peaks selected by DUBStepR, d-f) Monocle-generated trajectories on the corresponding UMAP visualizations. g) Hematopoietic differentiation schematic as described in Buenrostro et al. CLP: Common Lymphoid Progenitor, CMP: Common Myeloid Progenitor, GMP: Granulocyte/Macrophage Progenitor, HSC: Haematopoietic Stem Cell, LMPP: Lymphoid-primed Multipotent Progenitor, MEP: Megakaryocyte/Erythroid Progenitor, Mono: Monocyte, MPP: Multipotent Progenitor

Minor comments:

1) More information about normalization and clustering approaches of single-cell RNA-seq and single-cell ATAC-seq based on the selected features should be provided.

We apologise for the lack of information regarding the normalization and clustering approaches used for scRNA-seq and scATAC-seq data based on the selected features. Note that only the newly introduced rheumatoid arthritis dataset involves a clustering step. The ground truths for all other datasets are either cell lines or FACS labels. We have modified **Supp. Note 2: Datasets** in the manuscript to explain our processing steps as detailed below.

“3 cell line datasets.

The H2228, H1975 and HCC827 cell lines were used to make these datasets. The 3 datasets were generated by sequencing these cell lines on CEL-Seq2, 10x Genomics and Drop-seq platforms (Tian et al., 2019). All datasets were processed using Seurat (v3.1.0) or SingleCellExperiment (v1.12.0). The 10x and Drop-seq datasets were normalized using the LogNormalize function in Seurat, with a scale factor of 10000 and a pseudo-count of 1, before log transformation. The CEL-Seq2 dataset was logCPM normalized i.e. its cells were normalized to a scale factor of 1000000, before log transformation with a pseudo-count of 1.

5 cell line datasets.

The H2228, H1975, HCC827, H838 and A549 cell lines were used to make these datasets. The 2 datasets were generated by sequencing these cell lines on CEL-Seq2 and 10x Genomics platforms (Tian et al., 2019). For the 5 cell line dataset sequenced using CEL-Seq2, we combined the data obtained from the 3 plates (p1, p2 and p3) into one single dataset. We further selected only those cells that the demultiplexing result provided predicted as single cells. Finally, the data was logCPM normalized i.e. its cells were normalized to a scale factor of 1000000, before log transformation with pseudo-count of 1. The 5 cell line dataset sequenced using 10x was normalized using the LogNormalize function in Seurat, with a scale factor of 10000 and a pseudo-count of 1, before log transformation.

CRC Cell Line dataset.

The FPKM data from these cell lines was downloaded from GEO (GSE81861). This data has already passed the QC metrics defined by the authors. After removing replicates, we included the remaining 460 cells in this dataset. The FPKM values were then log-transformed with a pseudocount of 1.

FACS PBMC dataset.

The dataset was downloaded from the 10x Genomics website, and 2,600 cells were sampled from CD14+Monocytes, B cells, Naive CD4+ T cells, Naive CD8+ T cells and NK cells each. The resulting 13,000 cells were subjected to quality control

measures. A cell would only be included in the dataset if it had greater than 300 but less than 2,000 detected genes, and had a mitochondrial rate (proportion of reads originating from mitochondrial genes) of less than 8%. Due to sparsity of reads in this dataset, our standard filtering of genes expressed in 5% of cells rendered less than 4,000 genes. Hence, for this dataset, we modified our filtering threshold to keep genes expressed in at least 100 cells (approximately 1% of cells). The scRNA-seq counts were then normalized using the LogNormalize function in Seurat, with a scale factor of 10000 and a pseudo-count of 1, before log transformation.

Dataset for benchmarking computational scalability.

The Mouse Organogenesis Cell Atlas dataset was used for benchmarking scalability. The raw counts matrix for the dataset was downloaded from the Seattle Organismal Molecular Atlases (SOMA) Data portal (Cao et al., 2019). The counts were first used to create a Seurat object, which was downsampled to roughly 10,000 and 100,000 cells using the subset function in Seurat (v3.1.0). Labels provided by Cao et al. identified 37 clusters in this dataset. To preserve the biological heterogeneity of the dataset (for a fair performance comparison), we selected a maximum of 300 cells per cluster and 3,000 cells per cluster for the 10k and 100k datasets respectively. Clusters having fewer than the maximum number of cells were selected in their entirety. Following this, the datasets were used to create SingleCellExperiment objects (Amezquita et al., 2020). Genes expressed in at least 1 cell were retained and the counts were log-normalized with a pseudocount of 1, using the logNormCounts function within the scuttleR package (McCarthy et al., 2017). The log-normalized matrix was used as the input for the various algorithms tested.

Rheumatoid arthritis dataset.

Data preprocessing. The raw data files were used to generate a Seurat object. Filtering was performed to remove cells having >10% mitochondrial rate (calculated using the PercentageFeatureSet function in Seurat). Further, only cells with unique feature counts between 200 and 2,500 were retained, resulting in 8,350 cells. The data was then log-normalised using the NormalizeData function in Seurat at default settings.

Isolating T & NK cells. The normalized data was then annotated using SingleR (Aran et al., 2019), with the Monaco et al. (2019) immune dataset as the reference. The SingleR function was run at default settings, using the log-normalized counts as input. 38 cells were pruned by SingleR, leaving a total of 8,312 annotated cells. Of these, cells annotated as CD4+ T cells, CD8+ T cells or NK cells were isolated.

DUBStepR analysis. Due to the lower expression of genes in the T and NK cell population, we modified DUBStepR's gene filtering threshold to filter out genes expressed in less than 1% of the cells. The Seurat package was used for downstream processing. First, the feature genes were zero-centered and scaled, and PCA was performed. The top 12 PCs were selected using the elbow plot of

variance explained by the PCs. Clustering was performed using the Louvain graph-based approach - using the FindNeighbors function with 12 PCs and FindClusters at default parameter settings. UMAP coordinates were computed using the cell embeddings in 12 PCs.

Other feature selection methods. All other feature selection methods (HVGDisp, HVGST, trendVar, devianceFS, M3DropDANB, GiniClust and HLG) were run at default settings. To showcase the result of HVGST, we used the Seurat package to cluster cells, as described above. Similar to the DUBStepR result, 12 PCs were used for both clustering and UMAP visualization.

Single-cell ATAC-sequencing dataset.

We obtained the scATAC-seq dataset of 2,034 human hematopoietic progenitors from Buenrostro et al. (2018), wherein 491,437 peaks had been selected from the bulk hematopoietic ATAC-seq dataset in the paper. All peaks that were not open in at least 100 cells were removed from the dataset. This resulted in a data matrix of 2,034 cells x 32,090 peaks. Finally, every cell was normalized to a scale factor of 1000 (read-count normalization) using the NormalizeData function in Seurat with the parameter: normalization.method = "RC", so as to account for differences in overall accessibility of the cells. This normalized data was fed into DUBStepR and FindVariableFeatures for feature selection. DUBStepR was run with no additional feature filtering, while FindVariableFeatures was run at default settings.

After feature selection, PCA was performed on all 3 datasets - (i) all peaks, (ii) the top 2,000 highly variable features, and (iii) 6,764 DUBStepR-selected peaks. 50 PCs were used to capture the variance in the dataset in both cases, and the output of PCA was used for UMAP visualization.

For pseudotemporal ordering using Monocle 3, the Seurat object with the UMAP coordinates was converted to a cell_data_set object. Cells were clustered in Monocle 3 in the UMAP space using the cluster_cells function, with the following parameters: resolution = 1e-3 and num_iter = 10. The graph topology was constructed using the learn_graph function at default parameter settings, and nodes in the graph were manually defined at the end of every edge or at the intersection of any 2 or more edges. Nodes were colored based on the most abundant progenitor cell type in the proximity of the node in UMAP space."

2) How many cells per lung adenocarcinoma cell line? Fig 1a cell type should be replaced by cell line.

Cell Line	Number of cells
A549	1237
H1975	429
H2228	744
H838	841
HCC827	571

Table R3.1. Number of cells per lung adenocarcinoma cell line.

3) *Are other methods that failed to scale up due to software failure or consume endless time and huge memory? Does the computation time of DUBStepR include both regression, Guilt-by-association, and UMAP?*

HLG, GiniClust and trendVar failed to scale up to a million cells because of huge memory requirements. In scRNA-seq, most new computational methods (including DUBStepR) perform their entire analysis using sparse matrices, while these methods require a dense representation of the expression matrix to perform feature selection. The computation time for DUBStepR includes both regression and guilt-by-association. UMAP visualization is not included as part of the feature selection benchmarking.

4) *The confidence interval across seven datasets can be shown in Figure 3a.* We thank the reviewer for bringing this to our notice. We have added 90% confidence intervals to the data in the figure below (**Supp. Fig. S4**).

Supp. Fig. S4: Mean scaled Silhouette Index of feature sets ranging from 50 to 4,000 features, with 90% confidence intervals.

5) In Figure 3b, it is recommended to use different colors for different datasets to show the datasets DUBStepR failed to be the rank 1, and which algorithm might supplement that situation

We thank the reviewer for this suggestion. We have updated Fig. 3 (**Fig. 4** in revised manuscript) as recommended, with different colors representing the different datasets, as depicted below.

Fig. 4. Benchmarking feature selection methods. a) Mean scaled Silhouette Index of feature sets ranging from 50 to 4,000 features. b) Rank distribution of feature selection methods. For each dataset, the 7 methods are ranked from 1 to 7 by their best SI across all feature set sizes. c) AUROC of DE gene detection. *: $p \leq 0.05$, **: $p \leq 0.01$, ***: $p \leq 0.001$.

6) Figure 4a-b Density index label could be figure title to avoid overlap with a scatterplot.

We have updated Fig. 4a,b (Fig. 5a,b in the revised manuscript) as shown below.

Fig. 5. a-b) UMAP visualizations of the 5 cell lines comparing the local neighbourhood density of feature spaces in (a) good feature selection versus (b) poor feature selection.

7) *Figure 5 as an overview of the workflow might be organized in the first section of the main text and Figure 1.*

We thank the reviewer for this suggestion, and have repositioned the figure describing the DUBStepR workflow as **Fig. 1**.

Reviewers' Comments:

Reviewer #1:

Remarks to the Author:

In the revised manuscript, the clarity and presentation of the method and results are significantly improved. The authors have sufficiently addressed all my concerns in the previous round of review. I think the manuscript is in good shape now.

Reviewer #2:

Remarks to the Author:

Thank you for addressing the comments. I recommend the manuscript for publication for the community to try it.

Reviewer #3:

Remarks to the Author:

Thank you for addressing and answering all my previous major and minor comments. The methods description of the manuscript is significantly improved to be clearly presented for reproducible research, and the implemented tool DUBStepR has been shown to be seamlessly integrated into the canonical single-cell analysis workflow such as Seurat, SingleCellExperiment, and Monocle. The results are better organized with sufficient main and supplementary materials and corrections, especially for the newly added results from the Rheumatoid arthritis patient sample single-cell RNA-seq dataset, in which DUBStepR obviously identifies some interesting rare cell subtypes in T and NK cells. This real dataset application promisingly validates and supports DUBStepR's advantage over other tools to discover novel biological perspectives for future studies involving complex disease samples. However, there are still few issues that need to be resolved.

response to major comment:

4) In the response and Fig. 3i, there are 20 "seed" genes selected, but only 18 "seed" genes are shown here in Fig.3j, please be consistent with the number of genes. For the interpretation of the step-wise regression, the author thought no matter how many regression steps, the performance can still be improved. This might not be true, the authors may consider leveraging the K-fold cross-validation to test the performance of the algorithm on the held-out cells for different feature sizes.

6) The most interesting part of this manuscript version is the newly-added dataset, authors may consider uploading the processed or raw dataset onto GEO. In Figure 6 and corresponding main text, there are no a,b,c labels, figure 6 legend "clustered->clusters". in Supp Fig7, how are the platelet-T/NK doublets identified? Is cells doublet identification supported by DUBStepR?

Reviewer Response Letter

Reviewer #1 (Remarks to the Author):

In the revised manuscript, the clarity and presentation of the method and results are significantly improved. The authors have sufficiently addressed all my concerns in the previous round of review. I think the manuscript is in good shape now.

Reviewer #2 (Remarks to the Author):

Thank you for addressing the comments. I recommend the manuscript for publication for the community to try it.

Reviewer #3 (Remarks to the Author):

Thank you for addressing and answering all my previous major and minor comments. The methods description of the manuscript is significantly improved to be clearly presented for reproducible research, and the implemented tool DUBStepR has been shown to be seamlessly integrated into the canonical single-cell analysis workflow such as Seurat, SingleCellExperiment, and Monocle. The results are better organized with sufficient main and supplementary materials and corrections, especially for the newly added results from the Rheumatoid arthritis patient sample single-cell RNA-seq dataset, in which DUBStepR obviously identifies some interesting rare cell subtypes in T and NK cells. This real dataset application promisingly validates and supports DUBStepR's advantage over other tools to discover novel biological perspectives for future studies involving complex disease samples. However, there are still few issues that need to be resolved.

response to major comment:

4) In the response and Fig. 3i, there are 20 "seed" genes selected, but only 18 "seed" genes are shown here in Fig.3j, please be consistent with the number of genes.

We apologize for this error. The correct number of "seed" genes is actually 21. We have updated **Figure 3** in the manuscript, and copied it below.

Figure 3. a) Gene-gene correlation matrix of candidate feature genes (high correlation range score). b-d) Residuals from stepwise regression on the gene-gene correlation matrix. e) UMAP visualization of cells in an optimal feature space, colored by cell line. f-h) Same UMAP, colored by expression of genes regressed out in the first 3 steps. i) Screen plot: variance in GGC matrix explained by the gene regressed out at each step. j) Standardized average expression of the final seed gene set in each of the 5 cell lines. r: Pearson correlation.

For the interpretation of the step-wise regression, the author thought no matter how many regression steps, the performance can still be improved. This might not be true, the authors may consider leveraging the K-fold cross-validation to test the performance of the algorithm on the held-out cells for different feature sizes.

We thank the reviewer for this suggestion. When we designed DUBStepR, we considered optimising the number of regression steps to directly obtain the optimal feature set. The K-fold cross-validation approach suggested by the reviewer would have been an excellent strategy in this situation. However, as depicted in **Supp. Table S1** (shown below), the performance of stepwise regression hit a ceiling after 30 steps. The elbow point did not change beyond 30 steps, indicating that the variance explained by the subsequent features was negligible. Thus, optimising stepwise regression would not lead to an improvement in feature selection performance.

For this reason, we implemented a two-step approach in DUBStepR - stepwise regression to determine the initial seed gene set, and guilt-by-association to add correlated genes and expand the feature set. Our benchmarking analysis shows that DUBStepR's Silhouette Index (SI) increases as the feature set expands beyond the initial seed gene set using guilt-by-association (**Figure 4a** below), thus demonstrating a consistent improvement in performance regardless of the number of regression steps.

Dataset	20 steps	30 steps	50 steps	100 steps
FACS_PBMC	13	13	13	13
CRC_Cell_Line	19	22	22	22
3cl_10x	18	22	22	15
3cl_dropseq	14	14	14	14
3cl_celseq	12	11	11	11
5cl_10x	18	21	21	21
5cl_celseq	18	18	18	18

Supp. Table S1. Location of elbow point when regression is explicitly calculated for 20, 30, 50 and 100 steps.

Fig. 4. Benchmarking feature selection methods. a) Mean scaled Silhouette Index of feature sets ranging from 50 to 4,000 features.

6) *The most interesting part of this manuscript version is the newly-added dataset, authors may consider uploading the processed or raw dataset onto GEO.*

We have uploaded the processed Seurat object of our in-house-generated rheumatoid arthritis PBMC dataset to Zenodo, a publicly accessible repository, at <https://doi.org/10.5281/zenodo.4072260>.

In Figure 6 and corresponding main text, there are no a,b,c labels, figure 6 legend "clustered->clusters".

Figure 6 has been updated with subfigure labels and we have corrected the legend. Please find it copied below.

Figure 6. Analysis of lymphocyte population from rheumatoid arthritis patient PBMCs. a) UMAP visualization of clusters identified by DUBStepR. b-h) UMAP visualization of cell clusters identified by DUBStepR using features selected by the 7 other feature selection methods; b) HVGDisp, c) HGVST, d) trendVar, e) devianceFS, f) M3DropDANB, g) GiniClust, and h) HLG.

in Supp Fig7, how are the platelet-T/NK doublets identified? Is cells doublet identification supported by DUBStepR?

Platelet-T/NK doublets are identified using the gene expression signatures of platelets (PPBP, PF4), T cells (CD3D, CCR7) and NK cells (KLRF1). As depicted in **Supp. Fig. S5** (shown below), the platelet-T cluster expresses both platelet and T cell markers, while the platelet-NK cluster expresses both platelet and NK cell markers.

While DUBStepR is not explicitly designed to detect doublets, its sensitivity to unique gene expression signatures makes it a robust solution for picking out rare doublet populations in a heterogeneous single-cell experiment.

Supp. Fig. S5. Feature selection on rheumatoid arthritis T and NK cells using DUBStepR. a) Feature plots demonstrating expression of markers. b) Expression of *C1orf56* and its correlated cell state markers.